# Offline changepoint localization using a matrix of conformal p-values

**Sanjit Dandapanthula**                                      *sanjitd@cmu.edu*
*Department of Statistics*
*Carnegie Mellon University*

**Aaditya Ramdas**                                          *aramdas@cmu.edu*
*Department of Statistics and Machine Learning Department*
*Carnegie Mellon University*

**Reviewed on OpenReview:** *https://openreview.net/forum?id=bo2WlznUOc*

## Abstract

Changepoint localization is the problem of estimating the index at which a change occurred in the data generating distribution of an ordered list of data, or declaring that no change occurred. We present the broadly applicable MCP algorithm, which uses a matrix of conformal p-values to produce a confidence interval for a (single) changepoint under the mild assumption that the pre-change and post-change distributions are each exchangeable. We prove a novel conformal Neyman-Pearson lemma, motivating practical classifier-based choices for our conformal score function. Finally, we exemplify the MCP algorithm on a variety of synthetic and real-world datasets, including using black-box pre-trained classifiers to detect changes in sequences of images, text, and accelerometer data.

## 1 Introduction

In offline changepoint localization, we are (informally) given some ordered list of data and are told that there may have been a change in the data generating distribution at some unknown index, called the *changepoint*. Suppose, for example, that the data is drawn independently from a density $f_0$ before the changepoint and is drawn independently from another density $f_1 \neq f_0$ post-change. Then, the goal of changepoint localization is to estimate the index at which the change occurred, or to declare that no change occurred. In fact, using the tools of conformal prediction, we are able to non-trivially localize the changepoint *without any further assumptions about $f_0$ and $f_1$*.

The problem of changepoint localization arises frequently in statistical practice. For example, consider a firm monitoring the quality of an airplane part (via some metric $X$) after each flight. Suppose that at some point the part breaks, leading to lower performance in some downstream metric. The company may analyze this data weekly, and they may ask "after which flight did the part break?". In this scenario, it may neither make sense for the company to make parametric assumptions about the distribution of $X$, nor about the nature of the change (in the mean, variance, modality, etc.). Our algorithm solves this problem by producing a confidence set for the changepoint under far fewer assumptions than prior work.

Now, we are ready to formally describe the offline changepoint localization problem in a more general setting. In all of the following, we will let $[K] = \{1, \ldots, K\}$ for $K \in \mathbb{N}$ and use $\mathcal{M}(S)$ to denote the set of probability measures over $S$. Furthermore, we use $\stackrel{d}{=}$ to denote equality in distribution. We are given a list of $\mathcal{X}$-valued random variables $(X_t)_{t=1}^n$ for some $n \in \mathbb{N}$. Furthermore, there exists an unknown changepoint $\xi \in [n]$ such that $(X_t)_{t=1}^{\xi} \sim \mathbb{P}_0$ and $(X_t)_{t=\xi+1}^n \sim \mathbb{P}_1$ are respectively sampled from the pre-change distribution $\mathbb{P}_0 \in \mathcal{M}(\mathcal{X}^{\xi})$ and the post-change distribution $\mathbb{P}_1 \in \mathcal{M}(\mathcal{X}^{n-\xi})$. We use $\xi = n$ to denote the case where no change occurs. Let $\mathbb{P} = \mathbb{P}_0 \times \mathbb{P}_1$; we assume that the pre-change data is independent of the post-change data.

We make the very general assumption that $\mathbb{P}_0$ is *exchangeable*. This means that $\mathbb{P}_0$ is invariant to permutations in the following sense: for any permutation $\pi : [\xi] \to [\xi]$, we have:

$$(X_1, \ldots, X_\xi) \stackrel{d}{=} (X_{\pi(1)}, \ldots, X_{\pi(\xi)}).$$

For instance, it could be the case that $\mathbb{P}_0$ corresponds to i.i.d. observations according to some cumulative distribution function $F_0$. Similarly, we assume that the post-change distribution $\mathbb{P}_1$ is exchangeable as well (which occurs if $\mathbb{P}_1$ corresponds to i.i.d. observations according to some cumulative distribution function $F_1$). We summarize our main contributions below.

- We present the MCP algorithm, a novel and widely applicable method which uses a matrix of conformal p-values to produce a confidence interval for a changepoint under the mild assumption that the pre-change and post-change distributions are each exchangeable.

- We show that the MCP algorithm is able to produce (finite-sample) valid confidence sets for the changepoint under (only) the assumption that the pre-change and post-change distributions are exchangeable.

- We describe methods for learning the conformal score function used in the MCP algorithm from the data, thereby increasing the power of our method. These are based on a novel "conformal Neyman-Pearson" type result.

- We demonstrate the MCP algorithm on a variety of synthetic and real-world datasets, including using black-box classifiers to localize changes in sequences of images or text. We show that the MCP algorithm is able to produce narrow confidence sets for the changepoint, even when the change is difficult to detect.

This work focuses on localizing a single changepoint; one possible extension (which we do not explore here) is to use a variant of our algorithm to localize multiple changepoints. We begin by reviewing prior work on changepoint localization and conformal approaches to change detection in Section 2. We introduce the MCP algorithm in Section 3. We then detail several of the subroutines of the MCP algorithm in Section 4 and prove the validity of the resulting confidence sets. Next, we discuss how the score function should be chosen in Section 5. Finally, we apply the MCP algorithm on a variety of synthetic and real-world datasets in Section 7 to demonstrate its effectiveness and flexibility.

## 2 Related work

**Changepoint analysis.** Here, we give a review of several classical methods from changepoint analysis; some further details can be found in Truong et al. (2020). Traditional changepoint analysis has largely been dominated by parametric approaches based on the generalized likelihood ratio; however, these parametric methods often require knowing the pre-change and post-change distributions to achieve optimality, and often only give point estimators without associated confidence sets.

Methods such as CUSUM (Page, 1955) and conformal martingales (Vovk et al., 2003) are used for quickest (sequential) changepoint detection. One cannot use them to get confidence sets for the location of the changepoint (sometimes called changepoint localization). In fact, there is only one very recent work of Saha & Ramdas (2025) that attempts to provide confidence sets for changepoints after stopping a sequential detection procedure, but they make parametric assumptions, as well as assume that the pre- and post-change distributions are in known non-overlapping classes. We do not need such assumptions, and methods in their preprint are not applicable to our assumption-lean setup.

**Conformal prediction for change *detection*.** The application of conformal prediction to changepoint problems is relatively recent compared to traditional approaches. Conformal prediction, as surveyed in Shafer & Vovk (2008), provides a framework for constructing valid prediction regions with distribution-free guarantees under minimal statistical assumptions. While originally developed for supervised learning tasks, conformal methods have been extended to changepoint detection.

In contrast to traditional approaches to quickest changepoint *detection* (detecting whether a change occurred) using conformal martingales (Vovk et al., 2003), our method allows for *localization* (precise estimation of where the changepoint occurred). Our main technical contribution is to leverage *conformal p-values* constructed online from the data in the conformal martingale framework; these p-values are known to be i.i.d. and distributed as $\text{Unif}(0, 1)$ if no change has occurred yet. By a two-sample test, we are then able to localize the changepoint. Furthermore, in contrast to other conformal approaches to changepoint analysis, we allow for the score function to be learned from the data and prove a novel conformal Neyman-Pearson lemma guiding the choice of these score functions.

Early work on connecting conformal prediction to changepoint detection focused primarily on testing for exchangeability violations rather than precise localization. Vovk et al. (2003) proposed a general approach for testing the exchangeability assumption in an online setting using conformal martingales, laying important groundwork for changepoint detection. The conformal martingale approach is a special case of the e-detector framework laid out in Shin et al. (2023), which provides an extremely general and powerful method for *sequential* change detection. While most prior work using conformal prediction for change detection is in the sequential (online) setting, here we study the *localization* problem in the offline setting; we would like a confidence interval for the changepoint.

Building on this foundation, Vovk (2021) and Vovk et al. (2021) introduced conformal test martingales specifically designed for changepoint detection; for them, the motivation was to determine when the data generating distribution changes and a predictive algorithm needs to be retrained. They introduced conformal versions of the CUSUM and Shiryaev-Roberts procedures, which control false alarm rates through martingale properties. However, the primary emphasis remained on detection rather than localization of the changepoint.

Volkhonskiy et al. (2017) developed a more computationally efficient conformal test martingale, called the inductive conformal martingale, for change detection. Furthermore, they provided conformity measures and betting functions tailored specifically for change detection. However, their method does not provide formal guarantees for the localization task. More recently, Nouretdinov et al. (2021) investigated conformal changepoint detection under the assumption that the data generating distribution is continuous, showing that the conformal martingale is statistically efficient but without addressing the question of confidence interval construction for the changepoint location.

By bridging the gap between conformal prediction and classical changepoint localization using two-sample testing, our approach opens new possibilities for reliable changepoint analysis in complex, high-dimensional data settings where classical methods fail.

**Changepoint localization**  There is previous work in changepoint localization specifically for a change in the mean, but they rely on unspecified hyperparameters or are only asymptotically valid. Furthermore, many interesting changepoints do not involve a mean change (e.g., they may involve a change in the variance, modality, etc.). In Verzelen et al. (2023), the authors provide a CI for the changepoint, but the confidence interval in their Proposition 4 has several unknown constants (only their existence is ensured), and they do not provide a method to explicitly compute them (which is perhaps why the paper has no experiments). As a result, we could not implement their CI in practice, even in their special case of mean change. The confidence set in Cho & Kirch (2022) works only for mean changes in univariate data, is only asymptotically valid, and relies on a computationally expensive bootstrap procedure.

The SMUCE estimator introduced in Frick et al. (2014) provides an asymptotically valid confidence set in exponential family regression models. The paper Xu et al. (2024) produces a confidence set for high-dimensional regression problems under many technical assumptions, and the result in Fotopoulos et al. (2010) can be used to construct asymptotic CIs for a Gaussian mean change. However, note that all of these methods lack finite-sample validity and operate under far more stringent assumptions than our MCP algorithm.

Carlstein (1988); Lee (1996); Zou et al. (2014) study general nonparametric changepoint problems, but only provide consistent point estimators of the changepoint. Dümbgen (1991) provides a method to produce confidence sets in the general nonparametric setting, but this method is only asymptotically valid, requires a computationally expensive bootstrap procedure, and relies on several complex theoretical assumptions

which are difficult to verify in practice. When instantiated to an exponential family model, the method of Dümbgen (1991) provides an explicit formula for the confidence set matching that of Siegmund (1988).

**Nonparametric changepoint *testing*** Rank-based nonparametric tests, such as those developed by Pettitt (1979) and Ross & Adams (2012), offer distribution-free tests about *whether* there was a changepoint in a given dataset, but these methods do not produce a confidence set for the changepoint on rejection and suffer from lack of statistical power without additional assumptions. In contrast, MCP addresses these limitations by allowing the score function to be learned in a way that achieves nontrivial power, while also providing finite-sample valid confidence sets for changepoint location under minimal distributional assumptions.

**Multiple changepoint analysis** Existing methods for multiple changepoint analysis (which typically only yield point estimators) often rely on the so-called "Isolate-Detect" paradigm, using a sliding window or segmentation to reduce to the single changepoint case (for instance, see Section 5.2 of Truong et al. (2020) or Anastasiou & Fryzlewicz (2022)). In this sense, our algorithm is an important contribution to the changepoint analysis literature, since (after isolation of the changepoint) it provides a method for producing a confidence set.

## 3   The matrix of conformal p-values (MCP) algorithm

In this section, we discuss our MCP algorithm for conformal changepoint localization in technical detail. We begin by defining score functions; for more details on conformal prediction, see Shafer & Vovk (2008). In the following definition, we use $[\![\mathcal{X}^m]\!]$ to denote the set of unordered *bags* of $m$ data points (which may contain repetitions); such an unordered bag is often called a *multiset*. We will denote an element of $[\![\mathcal{X}^m]\!]$ by $[\![Y_1, \ldots, Y_m]\!]$, where the $Y_i$ are $\mathcal{X}$-valued random variables.

**Definition 3.1** (Score functions). A *family of score functions* is a list $((s_{rt})_{r=1}^n)_{t=1}^n$ of functions $s_{rt} : \mathcal{X} \times [\![\mathcal{X}^r]\!] \times \mathcal{X}^{n-t} \to \mathbb{R}$. Each element in a family of score functions is called a *score function*.

Note that a score function is intuitively intended to be a pre-processing transformation intended to separate the pre-change and post-change data points by projecting them into one dimension. In particular, the score function can be learned in any way that uses its second argument exchangeably, while its third argument can be used non-exchangeably. For intuition, we provide several examples of valid score functions in Appendix I and at the end of Section 5. The goal of our algorithms is to simultaneously test the null hypotheses that a change occurs at time $t \in [n]$:

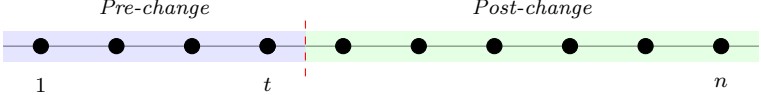

Figure 1: $\mathcal{H}_{0t} : \xi = t$ states that $(X_k)_{k=1}^t \sim \mathbb{P}_0$ and $(X_k)_{k=t+1}^n \sim \mathbb{P}_1$.

Then, the associated alternative hypotheses are of the form $\mathcal{H}_{1t} : \xi \neq t$. Fix $\alpha \in (0, 1)$. Ultimately, our algorithms will use p-values $p_t$ for $\mathcal{H}_{0t}$ to construct a $1 - \alpha$ confidence set for the changepoint:

$$\mathcal{C}_{1-\alpha} = \{t \in [n] : p_t > \alpha\},$$

If $n$ is present in the resulting confidence set, the practitioner should interpret this to mean that there was possibly no changepoint present in the dataset. On the other hand, if a point estimator for the changepoint is desired, we can output the estimate $\hat{\xi} = \arg\max_{t \in [n]} p_t$.

As described in Section 1, we assume that $\mathbb{P}_0$ and $\mathbb{P}_1$ are exchangeable. Then, we have the following algorithm for changepoint localization, which we call the MCP (matrix of conformal p-values) algorithm. In the following algorithm, let $u$ denote the cumulative distribution of a Unif$(0, 1)$ distribution. Furthermore, recall that the Kolmogorov-Smirnov (KS) distance between two cumulative distributions $F$ and $G$ is given by $\mathrm{KS}(F, G) = \sup_{z \in \mathbb{R}} |F(z) - G(z)|$. Our algorithm is motivated by the classical Kolmogorov-Smirnov test

for goodness of fit. Note that our algorithm does not specially depend on the choice of the KS distance; variants are in Appendix D.

---

**Algorithm 1:** MCP: matrix of conformal p-values

---

**Input:** $(X_t)_{t=1}^n$ (dataset), $((s_{rt}^{(0)})_{r=1}^n)_{t=1}^n$ and $((s_{rt}^{(1)})_{r=1}^n)_{t=1}^n$ (left and right score function families)
**Output:** $\mathcal{C}_{1-\alpha}$ (a level $1-\alpha$ confidence set for $\xi$)

**1** **for** $t \in [n]$ **do**
**2**    **for** $r$ *in* $(1,\dots,t)$ **do**
**3**      **for** $j$ *in* $(1,\dots,r)$ **do**
**4**       $\kappa_{rj}^{(t)} \leftarrow s_{rt}^{(0)}(X_j\,;\,[\![X_1,\dots,X_r]\!],(X_{t+1},\dots,X_n))$
**5**      **end**
     // compute per-anomaly p-values
**6**      $p_r^{(t)} \leftarrow \frac{1}{r}\sum_{j=1}^r \left(\mathbf{1}_{\kappa_{rj}^{(t)}>\kappa_{rr}^{(t)}} + \theta_r^{(t)}\mathbf{1}_{\kappa_{rj}^{(t)}=\kappa_{rr}^{(t)}}\right),$      where $\theta_r^{(t)} \sim \mathrm{Unif}(0,1)$
**7**    **end**
**8**    **for** $r$ *in* $(n,\dots,t+1)$ **do**
**9**      **for** $j$ *in* $(r,\dots,n)$ **do**
**10**       $\kappa_{rj}^{(t)} \leftarrow s_{n-r,\,n-t}^{(1)}(X_j\,;\,[\![X_{r+1},\dots,X_n]\!],(X_1,\dots,X_t))$
**11**      **end**
     // compute per-anomaly p-values
**12**      $p_r^{(t)} \leftarrow \frac{1}{n-r+1}\sum_{j=r}^n \left(\mathbf{1}_{\kappa_{rj}^{(t)}>\kappa_{rr}^{(t)}} + \theta_r^{(t)}\mathbf{1}_{\kappa_{rj}^{(t)}=\kappa_{rr}^{(t)}}\right),$      where $\theta_r^{(t)} \sim \mathrm{Unif}(0,1)$
**13**    **end**
**14**    $\hat{F}_0(z) := \frac{1}{t}\sum_{r=1}^t \mathbf{1}_{p_r^{(t)}\le z},$      $\hat{F}_1(z) := \frac{1}{n-t}\sum_{r=t+1}^n \mathbf{1}_{p_r^{(t)}\le z}$
**15**    $W_t^{(0)} \leftarrow \sqrt{t}\,\mathrm{KS}(\hat{F}_0,u),$      $W_t^{(1)} \leftarrow \sqrt{n-t}\,\mathrm{KS}(\hat{F}_1,u)$
**16**    Use Algorithm 2 to map $(W_t^{(0)},W_t^{(1)})$ to left and right p-values $(p_t^{\mathrm{left}},p_t^{\mathrm{right}})$
**17**    Combine $p_t^{\mathrm{left}}$ and $p_t^{\mathrm{right}}$ into a per-candidate p-value $p_t$ (Section 4.2)
**18** **end**
**19** $\mathcal{C}_{1-\alpha} \leftarrow \{t \in [n] : p_t > \alpha\}$
**20** **return** $\mathcal{C}_{1-\alpha}$

---

Before providing further intuition for MCP, we first note that the case $t = n$ may proceed slightly differently in order to improve power.

*Remark.* For the computation of $p_n$, we also recommend to compute the sequential ranks in reverse by

$$\overline{p}_t = \frac{1}{t}\sum_{j=n-t+1}^n \left(\mathbf{1}_{s(X_j)>s(X_t)} + \theta_t\,\mathbf{1}_{s(X_j)=s(X_t)}\right)$$

and calculate a "backward p-value" using a one-sample KS test from uniform using $(\overline{p}_1,\dots,\overline{p}_n)$. Then, the forward and backward p-values obtained from the respective KS tests can be combined using the Bonferroni method (Section 4.2). If the practitioner knows ahead of time that a changepoint exists in the dataset, the test for $\mathcal{H}_{0t}$ may be skipped in Algorithm 1 and the validity guarantee continues to hold.

Algorithm 1 effectively computes all of the *per-anomaly p-values* $p_r^{(t)}$, which can be written as an $n \times n$ matrix; we call this the *matrix of conformal p-values (MCP)*:

$$\mathrm{MCP} := \begin{bmatrix} p_1^{(1)} & p_1^{(2)} & \cdots & p_1^{(n)} \\ p_2^{(1)} & p_2^{(2)} & \cdots & p_2^{(n)} \\ \vdots & \vdots & \vdots & \vdots \\ p_n^{(1)} & p_n^{(2)} & \cdots & p_n^{(n)} \end{bmatrix}.$$

For each candidate changepoint $t \in [n]$ we construct $n$ per-anomaly p-values for $\mathcal{H}_{0t}$ (corresponding to a column in the MCP), where each per-anomaly p-value corresponds to one of the data points. Recall from

Algorithm 1 that these p-values are constructed by considering the randomized rank of that point's score, relative to the other points on the same side. Focusing on the $t$th column of the MCP, we then refine these p-values into left and right p-values as in Figure 2. Lastly, $p_t^{\text{left}}$ and $p_t^{\text{right}}$ are combined into the *per-candidate* p-value $p_t$, which we use to perform a hypothesis test of $\mathcal{H}_{0t} : \xi = t$ for a candidate changepoint. The computational complexity of the MCP algorithm is dependent on the choice of score function, and the base algorithm runs in $O(n^2)$ time. In fact, most of the score functions that we suggest in Section 5 and Appendix I can be computed with an $O(n^2)$ pre-processing step, so our algorithm typically has an overall quadratic complexity.

**Refinement of p-values in column $t$ of the MCP matrix**

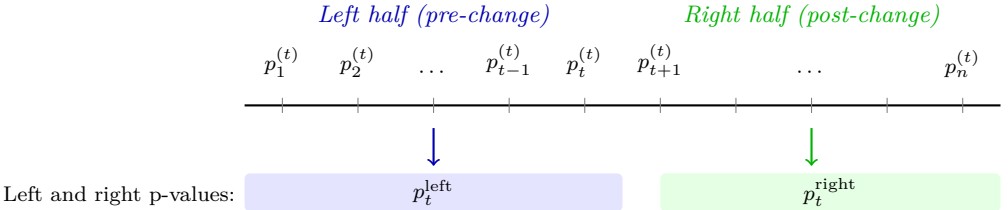

Figure 2: Refinement of the per-anomaly p-values in the $t$th row of the matrix of conformal p-values (MCP) into left and right p-values.

Here is some further intuition for the MCP algorithm (Algorithm 1). First, the algorithm splits the data into left and right halves at $t \in [n]$; we focus on the left half, since the right half is analogous. The goal of MCP is first to create p-values $(p_r^{(t)})_{r=1}^t$ which are independent. For each $r \in [t]$, we map the data points $(X_j)_{j=1}^r$ through a score function into exchangeable one-dimensional statistics $(\kappa_{rj}^{(t)})_{j=1}^r$ (line 4). Then, the (normalized) rank statistics of the scores are sequentially computed on the left and right halves (line 6). If there was no change in the left segment, the ranks for that segment would each be distributed like $\text{Unif}(0,1)$. In particular, as $t \to \infty$, we know by the Glivenko-Cantelli theorem that $\sup_{z \in \mathbb{R}} |\hat{F}_0(z) - u(z)| \to 0$.

Note that because $W_t^{(0)}$ only depends on the per-anomaly p-value $p_r^{(t)}$ for $r \in [t]$, it is a distribution-free statistic under $\mathcal{H}_{0t}$ and we can use it for hypothesis testing. Essentially, $W_t^{(0)}$ measures how far the normalized ranks in the left and right halves of the ordered list are from being uniform. We can convert this statistic to a p-value by a hypothesis test (line 16) which can be combined with the p-value obtained from the right half for each $t$ (line 18). Finally, we compute $p_n$ by testing for exchangeability as in the remark following Algorithm 1 and the tests can simultaneously be inverted to form a confidence set by the Neyman construction (line 19).

To motivate the normalization by $\sqrt{t}$ in the definition of $W_t^{(0)}$, recall that we have samples $p_r^{(t)} \sim \text{Unif}(0,1)$ for $r \in [t]$ and let $(\varphi_z)_{z \geq 0}$ denote the *Brownian bridge* $(\varphi_z = B_z - zB_1$, where $(B_z)_{z \geq 0}$ is a standard Brownian motion on $\mathbb{R}$). Then, a central limit theorem by Donsker (1951) implies that $\sqrt{t}\left(\sup_{z \in \mathbb{R}} |\hat{F}_0(z) - u(z)|\right) = \sqrt{t}\,\text{KS}(\hat{F}_0, u) \xrightarrow{d} \sup_{z \in [0,1]} |\varphi_z|$. Notice that normalizing by $\sqrt{t}$ stabilizes the KS distance, and we have an explicit form for the asymptotic distribution.

Under $\mathcal{H}_{0t}$, the scores $(\kappa_{rj}^{(t)})_{j=1}^r$ are exchangeable for $r \leq t$, meaning that the per-anomaly p-values are independent and uniform both to the left and right of $t$. We then aggregate the per-anomaly p-values to create the per-candidate p-value $p_t$, which allows us to test $\mathcal{H}_{0t}$. In Section 4, we discuss the subroutines of the MCP algorithm in greater depth. Note that Algorithm 1 is also symmetric under reflections of the data; if we reversed the input data as $\tilde{X} := (X_n, \ldots, X_1)$, the MCP algorithm would output $\tilde{\mathcal{C}}_{1-\alpha} := n - \mathcal{C}_{1-\alpha}$, where $\mathcal{C}_{1-\alpha}$ is the output of Algorithm 1 on $(X_1, \ldots, X_n)$.

# 4 Subroutines of the MCP algorithm and theoretical guarantees

In this section, we describe several key subroutines used in Algorithm 1 and prove its theoretical validity. We begin by discussing how to generate left and right p-values from the matrix of conformal p-values in Algorithm 1.

## 4.1 Hypothesis testing: generating left and right p-values

We now discuss how we can consolidate the matrix of p-values from the MCP algorithm into two p-values for each $\mathcal{H}_{0t}$, which we call the *left* and *right* p-values. This consolidation can be done using an empirical test. We can use simulations to construct confidence sets for the changepoint by estimating the level $1 - \alpha$ quantile of $W_t^{(0)}$ and $W_t^{(1)}$ under the null hypothesis that there is no changepoint. We use Algorithm 2 to construct left and right p-values. We also detail an alternative method based on an asymptotic KS test or permutation test in Appendix D, which can be used to speed up our algorithm when $t$ and $n - t$ are large. Performing the empirical test technically adds $O(Bn)$ to the time complexity, but is typically very fast in practice due to existence of pre-computed quantiles for the KS test in software packages.

---

**Algorithm 2:** Empirical test

**Input:** $(W_t^{(0)}, W_t^{(1)})$ (discrepancy scores), $B \in \mathbb{N}$ (user-chosen sample size)
**Output:** $(p_t^{\text{left}}, p_t^{\text{right}})$ (a pair of p-values for the left and right data)

**1 for** $b \in [B]$ **do**
**2** $\quad (X_t^{(b)})_{t=1}^n \sim \text{Unif}([0,1]^n)$
**3** $\quad$ Compute $(W_t^{(0,b)})_{t=1}^n$ and $(W_t^{(1,b)})_{t=1}^n$ using line 1 to line 15 of Algorithm 1 on simulated data $(X_t^{(b)})_{t=1}^n$
**4 end**
**5** Draw $\theta_0, \theta_1 \sim \text{Unif}(0,1)$
**6** $p_t^{\text{left}} \leftarrow \frac{1}{B+1} \left( \theta_0 + \sum_{b=1}^B (\mathbf{1}_{W_t^{(0,b)} > W_t^{(0)}} + \theta_0 \, \mathbf{1}_{W_t^{(0,b)} = W_t^{(0)}}) \right)$
**7** $p_t^{\text{right}} \leftarrow \frac{1}{B+1} \left( \theta_1 + \sum_{b=1}^B (\mathbf{1}_{W_t^{(1,b)} > W_t^{(1)}} + \theta_1 \, \mathbf{1}_{W_t^{(1,b)} = W_t^{(1)}}) \right)$
**8 return** $(p_t^{\text{left}}, p_t^{\text{right}})$

---

## 4.2 Combining p-values and constructing confidence sets

We now discuss how we can combine the left and right p-values into a single p-value $p_t$ for $\mathcal{H}_{0t}$. At the changepoint, we expect $p_\xi$ to be exactly uniformly distributed. Away from the changepoint, we would expect either the left or right p-value to be small.

Notice that the left and right p-values are independent if the score functions $s_{rt}^{(0)}$ and $s_{rt}^{(1)}$ have no dependence on its last argument; we call such score functions *non-adaptive*. Intuitively, a non-adaptive left score function $s_{rt}^{(0)}$ is one which does not use data to the right of $t$. On the other hand, it's clear that the left and right p-values won't necessarily be independent if the score functions are *adaptive* ($s_{rt}^{(0)}$ or $s_{rt}^{(1)}$ depend nontrivially on their last argument).

- **Minimum (requires independence):** The choice $p_t = 1 - (1 - \min\{p_t^{\text{left}}, p_t^{\text{right}}\})^2$ is a powerful combining method for change detection, since away from the changepoint, we would expect either the left or right p-value to be small. Under the null hypothesis $\mathcal{H}_{0t}$, we would expect $p_t^{\text{left}}$ and $p_t^{\text{right}}$ to be independent uniform random variables, so that $p_t$ is distribution-free and uniformly distributed. Note that one can use Fisher's method (Fisher (1928)) given by $p_t = F_{\chi_4^2}(-2\log(p_t^{\text{left}}) - 2\log(p_t^{\text{right}}))$; however, we have observed the minimum method to work better in practice.

- **Bonferroni correction (arbitrary dependence):** If the left and right p-values are dependent, then we can use the Bonferroni correction to combine the p-values, given by $p_t =$

$\min\{2p_t^{\text{left}}, 2p_t^{\text{right}}, 1\}$. This p-value is not distribution-free, but it is stochastically larger than uniform under the null hypothesis so we can use it for testing.

In practice, we have found the minimum and Bonferroni correction to be a good choice for combining function, under independence and dependence respectively.

### 4.3 Theoretical results

We begin with the following finite-sample coverage guarantee for the MCP algorithm (Algorithm 1).

**Theorem 4.1** (Coverage guarantee (empirical test)). *Suppose that $(X_t)_{t=1}^n$ is an exchangeable sequence of random variables with a changepoint $\xi \in [n]$. Let $\mathcal{C}_{1-\alpha}$ be the level $1-\alpha$ confidence set constructed with Algorithm 1, using the empirical test described in Section 4.1 to construct left and right p-values and using any algorithm in Section 4.2 to combine these p-values. Then, the coverage of $\mathcal{C}_{1-\alpha}$ is at least $1-\alpha$:*

$$\mathbb{P}\left(\xi \in \mathcal{C}_{1-\alpha}\right) \geq 1 - \alpha.$$

In particular, note that if there is no changepoint, then $\xi = n$, and the confidence set will contain $n$ with probability at least $1-\alpha$. We will see in experiments that when $\xi \neq n$, the confidence sets typically omit $n$.

Recently, Bhattacharya & Ramdas (2025) showed that in the parametric setting and under relatively general triangular array assumptions, the relative length of the confidence set produced by the MCP algorithm converges to zero as $n \to \infty$. Under the same assumptions, that paper also showed consistency of the point estimator $\hat{\xi} := \arg\max_{t \in [n]} p_t$ to the true changepoint $\xi$ at the rate $o_p(n^{1/4})$.

## 5 How should the score function be chosen?

The score function in the MCP algorithm should be chosen to maximize the power of the test. In this section, we will discuss how the score function should be chosen in practice.

We begin by proving a Neyman-Pearson type lemma for conformal prediction. Suppose we observe independent $\mathcal{X}$-valued data $X_1, \ldots, X_{n+1}$ and we wish to test the null $\mathcal{H}_0 : X_1, \ldots, X_{n+1}$ are i.i.d. against the alternative $\mathcal{H}_1 : X_1, \ldots, X_k$ are i.i.d. but $X_{k+1}, \ldots, X_{n+1}$ are i.i.d. from a different distribution, for some $k \in [n]$. Suppose further that we are forced to use a conformal p-value to perform this test. To elaborate, we must choose a non-conformity *score function* $s : \mathcal{X} \to \mathbb{R}$, draw $\theta_1, \ldots, \theta_{n+1} \sim \text{Unif}(0,1)$ i.i.d., calculate the p-value

$$p_n[s] = \frac{1}{n+1} \sum_{i=1}^{n+1} (\mathbf{1}_{s(X_i) > s(X_{n+1})} + \theta_i \, \mathbf{1}_{s(X_i) = s(X_{n+1})}),$$

and reject the null at level $\alpha$ when $p_n[s] \leq \alpha$.

The natural question then is: what is the optimal (oracle) score function for this task? Let $Q$ denote the distribution of $X_i$ for $i \in \{k+1, \ldots, n+1\}$ and let $R$ denote the distribution of $X_i$ for $i \in [k]$. Defining the densities $q = dQ/d(Q+R)$ and $r = dR/d(Q+R)$, the answer to the above question is the likelihood ratio $q(x)/r(x)$, as we will formalize below.

It will be easier to state the result in terms of the normalized rank functional

$$T_n[s] = \frac{1}{n} \sum_{i=1}^{n} (\mathbf{1}_{s(X_i) < s(X_{n+1})} + \theta_i \, \mathbf{1}_{s(X_i) = s(X_{n+1})}),$$

where $\theta_1, \ldots, \theta_{n+1} \sim \text{Unif}(0,1)$ are i.i.d. and independent of everything else. Then, we reject $\mathcal{H}_0$ if $T_n[s] \geq t_{n,1-\alpha}$ for some threshold $t_{n,1-\alpha}$.

**Theorem 5.1** (Conformal Neyman-Pearson lemma). *Let $s^*(x) = q(x)/r(x)$ denote the likelihood ratio non-conformity score. Then, for any other $s$, we have the optimality result*

$$\mathbb{E}[T_n[s^*]] \geq \mathbb{E}[T_n[s]].$$

*Define the test $\phi_\alpha[s] \coloneqq \mathbf{1}_{T_n[s] \geq t_{n,1-\alpha}}$ for some fixed threshold $t_{n,1-\alpha} \geq 0$ and let $\alpha$ denote the level of $\phi_\alpha[s^*]$ ($\alpha = \mathbb{E}[\phi_\alpha[s^*]]$ when $Q = R$). Then for any level $\alpha$ test $\phi$ of $\mathcal{H}_0$ against $\mathcal{H}_1$, it holds that*

$$\mathbb{E}[\phi_\alpha[s^*]] \geq \mathbb{E}[\phi].$$

*In particular, it follows that*

$$\mathbb{E}[\phi_\alpha[s^*]] \geq \mathbb{E}[\phi_\alpha[s]]$$

*for any choice of $s$.*

Note that the proof of Theorem 5.1 (given in Appendix F) also shows a stronger result; define the conditional power

$$\beta_s(u) = \mathbb{E}\left[ T_n[s] \ \middle| \ s(X_{n+1}) = F_{s(X_{n+1})}^{-1}(u) \right],$$

which exists and is unique for Lebesgue-almost every $u \in (0,1)$ by the disintegration of measure. Then, we have that (for Lebesgue-almost every $u \in (0,1)$)

$$\beta_s(u) \leq \beta_{s^*}(u).$$

In this sense, the likelihood ratio score yields the uniformly most powerful conformal test. A similar result can be shown without randomizing using the $\theta_i$, assuming that the scores are almost surely distinct. Appendix G presents an analogous result for e-values.

These results motivate the use of likelihood ratio type score functions in our experiments. Indeed, even when the likelihood ratio is not known exactly, one can hope to learn it and approximate the optimal tests. In particular, we can use a pre-trained multi-class classifier to create a score function, which allows our algorithm to form narrow confidence sets in extremely general settings. Suppose we have a pre-trained multi-class classifier $\hat{g} : \mathcal{X} \to \Delta^{\mathcal{S}}$, where $\mathcal{S}$ is a discrete set of labels and $\Delta^{\mathcal{S}}$ represents the probability simplex over $\mathcal{S}$:

$$\Delta^{\mathcal{S}} = \left\{ p \in [0,1]^{|\mathcal{S}|} : \sum_{s \in \mathcal{S}} p_s = 1 \right\}.$$

Furthermore, assume that $f_0$ is a density over $g^{-1}(s_0)$ for some $s_0 \in \mathcal{S}$, where $g$ is the ground truth classifier. Similarly, assume that $f_1$ is a density over $g^{-1}(s_1)$ for some $s_1 \in \mathcal{S}$. We can use the most popular class index

$$\hat{s}(z_1, \ldots, z_k) = \arg\max_{s \in \mathcal{S}} |\{1 \leq i \leq k : \hat{g}(z_i)_s \geq \hat{g}(z_i)_{s'} \text{ for all } s' \in \mathcal{S}\}|.$$

to estimate the likelihood ratio score function:

$$s_{rt}(x_r; [\![x_1, \ldots, x_r]\!], (x_{t+1}, \ldots, x_n)) = \frac{\hat{g}(x_r)_{\hat{s}(x_1, \ldots, x_r)}}{\hat{g}(x_r)_{\hat{s}(x_{t+1}, \ldots, x_n)}}.$$

By precomputing the most common class for the prefix and postfix arrays, this score function does not incur any additional computational cost over using the oracle likelihood ratio score function. We give several additional methods for choosing the score function in Appendix I. Finally, we demonstrate the performance of our algorithms through simulations.

## 6 Extension to multiple changepoints

In this section, we describe how the MCP algorithm (Algorithm 1) can be used to localize multiple changepoints. Suppose we are interested in a model where $(X_{\xi_k+1}, \ldots, X_{\xi_{k+1}}) \sim \mathbb{P}_k$ for $0 \leq k \leq K$, where $0 = \xi_0 < \xi_1 < \cdots < \xi_{K+1} = n$. Since we would like to apply the MCP algorithm, we assume that $\mathbb{P}_k$ is

exchangeable for all $0 \leq k \leq K$ (the data distribution between changepoints is exchangeable). Now, we would like to recover a confidence set which covers all of the changepoints $\{\xi_k\}_{k=1}^{K}$ with high probability.

To achieve this goal, we consider the popular kernel changepoint detection (KCPD) framework for finding point estimators of multiple changepoints (Harchaoui & Cappé, 2007; Arlot et al., 2019). Under mild detectability and minimum separation assumptions, denoting the minimum separation between by $\ell$ and fixing any $\epsilon > 0$, Diaz-Rodriguez & Jia (2025) recently showed that the KCPD estimator is consistent in the sense that the number of estimated changepoints is $K$ and the estimated changepoints are all $(\epsilon\ell)$-close to the true changepoints, with probability tending to one.

Now, suppose we are on the high-probability event where the estimated number of changepoints is $K$ and the estimated changepoints are all $(\ell/2)$-close to the true changepoints; denote the estimated changepoints by $(\hat{\xi}_k)_{k=1}^{K}$. Let $t_k := (\hat{\xi}_k + \hat{\xi}_{k-1})/2$ for $k \in [K+1]$, defining $\hat{\xi}_0 := 0$ and $\hat{\xi}_{K+1} := n$. Then, the segments of data $(X_{t_k}, X_{t_{k+1}})$ have exactly one changepoint contained within them for $k \in [K]$, such that the data to the left and right of the true changepoint is exchangeable. Thus, the MCP algorithm applied separately to each of the intervals $(X_{t_k}, X_{t_{k+1}})$ produces a $1 - \alpha$ confidence set $\mathcal{C}_{1-\alpha}^{(k)}$ for $\xi_k$.

Putting the above considerations together, we can produce a heuristic asymptotic $1 - \alpha$ confidence set for $\xi_k$ using the MCP algorithm as a wrapper over the KCPD algorithm. We exemplify this method to localize multiple Gaussian mean changes. In this simulation, we choose $n = 1500$ with $K = 4$ and $(\xi_1, \xi_2, \xi_3, \xi_4) = (150, 500, 820, 1100)$. Here, we choose $\mathbb{P}_k = \mathcal{N}(\mu_k, 1)$ with $(\mu_0, \mu_1, \mu_2, \mu_3, \mu_4) = (-1, 1/2, 3/2, -2, -1)$. We depict the input data in Figure 3.

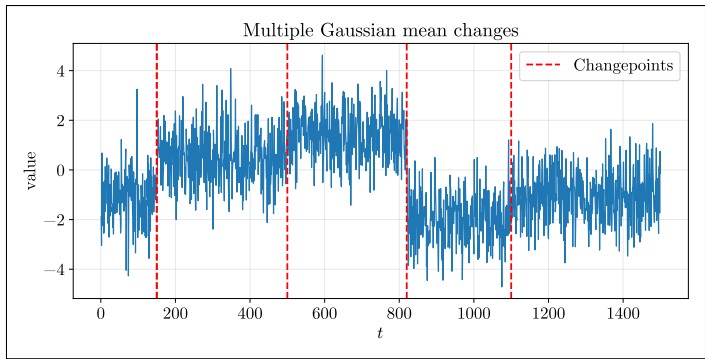

Figure 3: Sample of multiple Gaussian mean change data.

We apply the KCPD algorithm as described in Diaz-Rodriguez & Jia (2025) using the squared exponential kernel

$$k(z, z') = \exp\left(\frac{(z - z')^2}{2\sigma^2}\right),$$

where $\sigma > 0$ is chosen to be the median of $(|X_i - X_j|)_{1 \leq i < j \leq n}$. Then, we apply the MCP algorithm on the intervals $(X_{t_k}, X_{t_{k+1}})$ for $k \in [K]$ using the oracle likelihood ratio score function as described in Section 5; we depict the results in Figure 4.

The estimated changepoints from KCPD are $(150, 497, 820, 1091)$. Now the MCP confidence set for $\xi_1$ is $[111, 163]$, the confidence set for $\xi_2$ is $[444, 541]$, the confidence set for $\xi_3$ is $[807, 846]$, and the confidence set for $\xi_4$ is $[1072, 1163]$. In particular, once the KCPD algorithm isolates each changepoint, the MCP algorithm is able to produce confidence sets for each changepoint separately. This is an instance of the popular "Isolate-Detect" paradigm in changepoint localization (Anastasiou & Fryzlewicz, 2022), which consists of isolating each changepoint to lie in an interval and then applying a single-changepoint algorithm.

## 7  Experiments

We now demonstrate that the MCP algorithm can be used to construct narrow confidence sets for the changepoint in a variety of settings, and that our confidence sets have good empirical coverage and width.

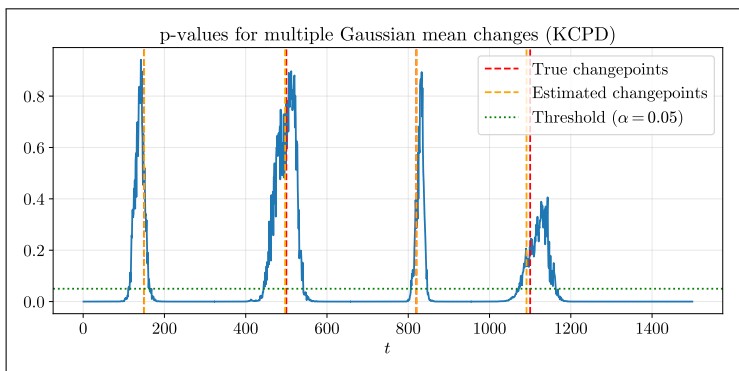

Figure 4: p-values for multiple Gaussian mean changes using MCP as a wrapper on KCPD. The dashed red lines indicate the true changepoints and the dashed yellow lines indicate the point estimators from KCPD. The regions on the horizontal axis where the p-values lie above the dotted green line ($\alpha = 0.05$) correspond to our heuristic 95% confidence sets for each changepoint.

All code for these experiments, including generic implementations of the MCP algorithm, can be found at the following GitHub repository:

$$\texttt{https://github.com/sanjitdp/conformal-change-localization}.$$

All experiments are run using the CPU on our personal MacBook Pro with an M1 Pro processor; our algorithm takes under 10 seconds to run in each experiment.

## 7.1 Gaussian mean change

We begin with a Gaussian mean change simulation. Suppose that we observe data of length $n = 1000$ and the changepoint is $\xi = 400$. Furthermore, suppose that $\mathbb{P}_0 = \otimes_{t=1}^{\xi} \mathcal{N}(-1, 1)$ and $\mathbb{P}_1 = \otimes_{t=\xi+1}^{n} \mathcal{N}(1, 1)$. First, as a baseline, suppose that we have the oracle likelihood ratio score function, as described in Section 5. We can then use the MCP algorithm to find discrepancy scores $((W_t^{(i)})_{i=0}^{1})_{t=1}^{n-1}$ and transform them into left and right p-values using Algorithm 2. Finally, we use the methods described in Section 4.2 to combine the left and right p-values into a single p-value for $\mathcal{H}_{0t}$ using the minimum combining function and construct a $1 - \alpha$ confidence set for the changepoint by inverting the test. We plot the resulting p-values in Figure 5a.

We begin by discussing our test for $\mathcal{H}_{0n}$ in Algorithm 1. When we test $\mathcal{H}_{0n}$ at level $\alpha_0 = 0.01$ using the identity score function and only the forward p-values, we estimate over 1000 simulations (with no changepoint) that the Type I error rate of our test for $\mathcal{H}_{0n}$ is 0.009, which aligns closely with the theoretical Type I error of $\alpha_0 = 0.01$. In these cases (when there is no changepoint), the resulting confidence set is sometimes much wider, but the above discussion shows that it usually correctly contains $n$; we expand on these findings in Appendix E. On the other hand, over 1000 simulations (with a changepoint), the test for $\mathcal{H}_{0n}$ was always able to detect a deviation from exchangeability and had an empirical power of 1.000.

In all future experiments, the dataset contains a changepoint and the test for $\mathcal{H}_{0n}$ correctly rejects the null hypothesis; therefore, $n$ is not included in the resulting confidence sets. Suppose that we do not have access to the oracle score function. Then, as described in Section 5, we can use a kernel density estimator to estimate the likelihood ratio using the data before and after $t$ (for each $t \in [n-1]$), using a Gaussian kernel. We choose bandwidth $10^{-1}$ when $t \in \{1, n-1\}$ and bandwidth $r^{-1/5}$ otherwise, where $r$ is the number of samples used to learn each KDE; this is known as Scott's rule (due to Scott (1979)). Then, we repeat the process described above to compute the left and right p-values. However, due to dependence of the left and right p-values, we use the Bonferroni combining rule, plotting the resulting p-values $(p_t)_{t=1}^{n-1}$ in Figure 5b.

We obtain the confidence set $[359, 435]$ using the oracle score function and the confidence set $[324, 424]$ using the learned score function. In particular, the width of our confidence interval does not suffer much (compared to the oracle likelihood ratio score function) even if our score is learned using a kernel density estimator.

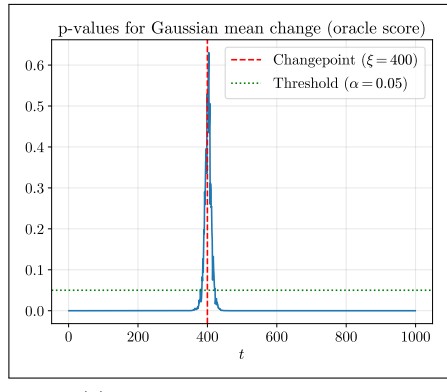 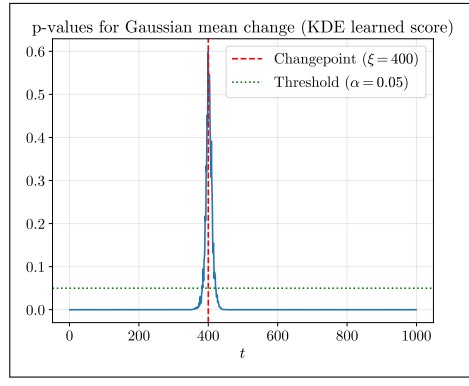

(a) oracle score function family          (b) KDE learned score function family

Figure 5: p-values for a Gaussian mean change at $\xi = 400$ (a single run). The dashed red line indicates the true changepoint, and the region on the horizontal axis where the p-values lie above the dotted green line ($\alpha = 0.05$) corresponds to our 95% confidence set. The point estimators are (a) $\hat{\xi} = 409$ and (b) $\hat{\xi} = 380$, both of which are close to the true $\xi = 400$.

Next, in Figure 6, we plot the resulting p-values when the true changepoint is closer to the edge of the dataset. We observe that the size of the confidence set and quality of the point estimator do not suffer too much even when the changepoint is very close to the edge of the dataset. By symmetry, we expect similar results to hold when the true changepoint is in the right half of the data.

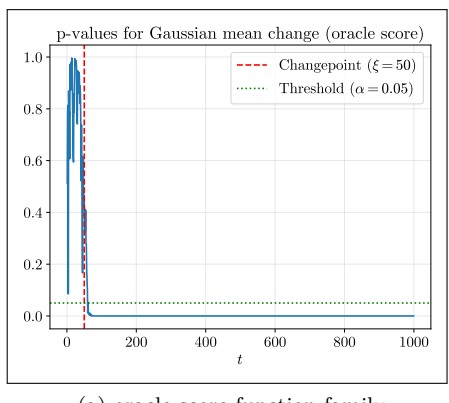 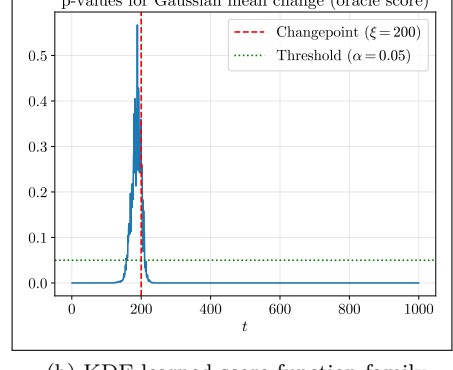

(a) oracle score function family          (b) KDE learned score function family

Figure 6: p-values for a Gaussian mean change at (a) $\xi = 50$ and (b) $\xi = 200$ (a single run). The dashed red line indicates the true changepoint, and the region on the horizontal axis where the p-values lie above the dotted green line ($\alpha = 0.05$) corresponds to our 95% confidence set. The point estimators are (a) $\hat{\xi} = 14$ and (b) $\hat{\xi} = 189$, both of which are relatively close to the true changepoints. The sizes of the confidence sets are (a) 60 and (b) 53.

### 7.1.1 Frequentist operating characteristics of MCP

Running the same simulation 200 times for a change from $\mathcal{N}(-1, 1)$ to $\mathcal{N}(1, 1)$ at $\xi = 400$ with $n = 1000$ samples, we plot the empirical power of our test for $\mathcal{H}_{0t}$ against various values of $t \neq \xi = 400$ in Figure 7. As expected, we observe that the power of the test increases as we move further away from the true changepoint.

Additionally, we report the empirical average width and coverage of our confidence sets at various levels, as well as the bias and mean absolute deviation of our point estimator $\hat{\xi} = \arg\max_{t \in [n]} p_t$, in Table 1. In Table 1, we also compare the performance of the MCP algorithm on datasets with varying coverage, signal strength and sample size.

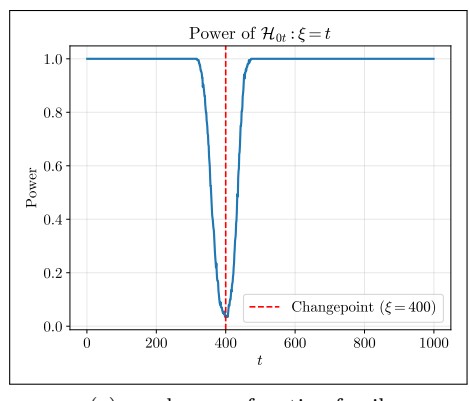 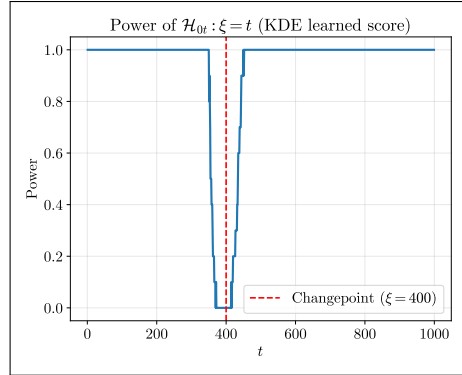

(a) oracle score function family        (b) KDE learned score function family

Figure 7: Empirical power of the MCP test for $\mathcal{H}_{0t}$ against various values of $t \neq \xi = 400$, using (a) the oracle likelihood ratio score function (averaged over 200 runs) and (b) the KDE score function (averaged over 10 runs).

| $n$ | $\delta$ | Target $\alpha$ | Avg. width | Empirical coverage | Bias | Mean abs. dev. |
|---|---|---|---|---|---|---|
| 100 | 2.0 | 0.05 | $22.29_{\text{se: }6.9}$ | $0.93_{\pm 0.03}$ | $-0.14_{\text{se: }4.7}$ | $3.50_{\text{se: }3.1}$ |
| | | 0.5 | $6.63_{\text{se: }4.9}$ | $0.50_{\pm 0.07}$ | | |
| | 1.0 | 0.05 | $31.41_{\text{se: }10.1}$ | $0.47_{\pm 0.03}$ | $-0.33_{\text{se: }6.6}$ | $5.10_{\text{se: }4.6}$ |
| | | 0.5 | $9.03_{\text{se: }6.4}$ | $0.94_{\pm 0.07}$ | | |
| 200 | 2.0 | 0.05 | $29.18_{\text{se: }8.0}$ | $0.96_{\pm 0.03}$ | $-0.50_{\text{se: }5.6}$ | $4.43_{\text{se: }3.6}$ |
| | | 0.5 | $8.88_{\text{se: }6.6}$ | $0.54_{\pm 0.07}$ | | |
| | 1.0 | 0.05 | $38.96_{\text{se: }11.7}$ | $0.97_{\pm 0.03}$ | $-0.12_{\text{se: }7.9}$ | $5.96_{\text{se: }5.4}$ |
| | | 0.5 | $12.12_{\text{se: }8.7}$ | $0.53_{\pm 0.07}$ | | |
| 500 | 2.0 | 0.05 | $41.05_{\text{se: }12.0}$ | $0.93_{\pm 0.03}$ | $-2.22_{\text{se: }8.7}$ | $7.02_{\text{se: }5.3}$ |
| | | 0.5 | $13.04_{\text{se: }10.3}$ | $0.53_{\pm 0.07}$ | | |
| | 1.0 | 0.05 | $55.13_{\text{se: }16.7}$ | $0.94_{\pm 0.03}$ | $-0.78_{\text{se: }12.6}$ | $8.96_{\text{se: }8.5}$ |
| | | 0.5 | $17.42_{\text{se: }13.3}$ | $0.50_{\pm 0.07}$ | | |

Table 1: Average width and coverage of confidence sets for Gaussian mean change, as well as the bias ($\mathbb{E}[\hat{\xi} - \xi]$) and mean absolute deviation ($\mathbb{E}[|\hat{\xi} - \xi|]$) of the point estimator over 1000 trials. The length of the data is $n$ and we observe a change from $\mathcal{N}(-\delta, 1)$ to $\mathcal{N}(\delta, 1)$ at $\xi = 2n/5$. We run the MCP algorithm using the oracle likelihood ratio score function to construct a $1 - \alpha$ confidence set for the changepoint. Note that the empirical coverage is very close to $1 - \alpha$, and the bias is very small relative to $n$ (though systematically negative, because the changepoint is in the left half). Binomial errors are reported for empirical coverage (95%) and standard errors are reported for all other values.

We know from Theorem 4.1 that the true coverage probability is at least 50% and 95% respectively for a 50% or 95% confidence interval constructed using our method; this is what we observe empirically as well. Furthermore, we observe that the point estimator is empirically close to the true changepoint; this is expected due to a recent result of Bhattacharya & Ramdas (2025) showing that the point estimator $\hat{\xi}$ is consistent at the rate $o_p(n^{1/4})$ in a general triangular array setup. Additionally, we observe in practice that because the changepoint is in the left half of the data, the point estimator tends to be biased towards the left of the true changepoint.

### 7.1.2 Comparison against baselines

Next, we compare the performance of the MCP algorithm against several baselines for changepoint localization. As discussed in Section 2, there are relatively few existing methods for constructing confidence sets for changepoints, especially in the nonparametric setting. We consider one baseline algorithm from Siegmund (1988), which constructs confidence intervals for the changepoint under a (parametric) exponential family assumption by deriving an approximation to the asymptotic distribution of the MLE. We also consider the bootstrap-based algorithm from Cho & Kirch (2022), which constructs confidence intervals for a nonparametric univariate mean change. We compare the average width and empirical coverage of our confidence sets against these baselines in Table 2. Here, we run all methods on data of length $n = 1000$ with a change from $\mathcal{N}(-1, 1)$ to $\mathcal{N}(1, 1)$ at $\xi = 400$.

| Method | Avg. width | Empirical coverage | Bias | Mean abs. dev. |
|---|---|---|---|---|
| MCP (ours) | $41.69_{\text{se: }0.6}$ | $0.96_{\pm 0.03}$ | $-0.32_{\text{se: }9.0}$ | $7.02_{\text{se: }5.6}$ |
| Siegmund (1988) | $1.31_{\text{se: }0.5}$ | $1.00_{\pm 0.03}$ | $-0.04_{\text{se: }0.3}$ | $0.15_{\text{se: }0.2}$ |
| Cho & Kirch (2022) | $3.00_{\text{se: }0.1}$ | $0.94_{\pm 0.03}$ | $1.04_{\text{se: }0.3}$ | $1.05_{\text{se: }0.3}$ |

Table 2: Average width and empirical coverage of confidence sets for Gaussian mean change, as well as the bias ($\mathbb{E}[\hat{\xi} - \xi]$) and mean absolute deviation ($\mathbb{E}[|\hat{\xi} - \xi|]$) of the point estimator over 1000 trials. The length of the data is $n = 1000$ and we observe a change from $\mathcal{N}(-1, 1)$ to $\mathcal{N}(1, 1)$ at $\xi = 400$. We run the MCP algorithm using the oracle likelihood ratio score function to construct a 95% confidence set for the changepoint, and compare against the methods of Siegmund (1988) and Cho & Kirch (2022). Binomial errors are reported for empirical coverage (95%) and standard errors are reported for all other values.

We observe from Table 2 that the methods of Siegmund (1988) and Cho & Kirch (2022) do produce sharper confidence sets under their respective modeling assumptions, but our MCP algorithm operates under far more general assumptions. For instance, our method does not require the existence of *any* moments of the data distribution, and is able to operate on high-dimensional data. To illustrate this point, we consider a change from a Cauchy distribution with location $-1$ to a Cauchy distribution with location 1 at $\xi = 400$. The method of Cho & Kirch (2022) is also not applicable here, since the Cauchy distribution does not have a finite mean, and Siegmund (1988) is misspecified here, since the data is not Gaussian. We show in Table 3 that these methods fare poorly, but our MCP algorithm is able to produce valid confidence sets even in this challenging scenario, even with misspecified score function (i.e. even if we incorrectly assumed the data was Gaussian to design the score, our coverage is maintained).

Furthermore, as we demonstrate in the following sections, our MCP algorithm is able to operate in many practical settings (e.g., text data, accelerometer data, and image data) where existing methods such as those of Siegmund (1988) and Cho & Kirch (2022) are not applicable.

### 7.1.3 Visualizing the matrix of conformal p-values

To give more intuition for the MCP algorithm, we can also visualize the matrix of conformal p-values (MCP) as described in Section 3, in Figure 8. We consider the setting where $n = 1000$ and there is a change from $\mathcal{N}(-1, 1)$ to $\mathcal{N}(1, 1)$ at $\xi = 400$. Observe that only when $t = \xi = 400$ are the p-values in row $t$ truly uniformly distributed and independent in the left and right halves respectively, as they should be under $\mathcal{H}_{0t}$. For instance, if $t < \xi$, then the right p-values will be invalid and the Kolmogorov-Smirnov statistic $W_t^{(1)}$ used in the MCP algorithm (Algorithm 1) will detect their deviation from uniformity.

| Method | Avg. width | Empirical coverage | Bias | Mean abs. dev. |
|---|---|---|---|---|
| MCP – oracle score (ours) | $53.27_{\text{se: }17.4}$ | $0.97_{\pm 0.03}$ | $0.25_{\text{se: }19.8}$ | $8.78_{\text{se: }10.9}$ |
| MCP – Gaussian score (ours) | $70.69_{\text{se: }20.1}$ | $0.94_{\pm 0.03}$ | $-3.685_{\text{se: }32.2}$ | $17.40_{\text{se: }20.3}$ |
| Siegmund (1988) | $1.01_{\text{se: }0.6}$ | $0.12_{\pm 0.03}$ | $31.79_{\text{se: }110.9}$ | $77.94_{\text{se: }84.5}$ |
| Cho & Kirch (2022) | $181.46_{\text{se: }48.1}$ | $0.82_{\pm 0.03}$ | $23.96_{\text{se: }67.7}$ | $50.00_{\text{se: }53.2}$ |

Table 3: Average width and empirical coverage of confidence sets for a Cauchy location change over 1000 trials. The length of the data is $n = 1000$ and we observe a change from $\text{Cauchy}(-1, 1)$ to $\text{Cauchy}(1, 1)$ at $\xi = 400$. We run the MCP algorithm using both the oracle likelihood ratio score function as well as the misspecified likelihood ratio score function assuming that there is a change from $\mathcal{N}(-1, 1)$ to $\mathcal{N}(1, 1)$, to construct a 95% confidence set for the changepoint. We compare against the method of Siegmund (1988), which produces narrow confidence sets with extremely low coverage, as well as the method of Cho & Kirch (2022), which produces wide confidence sets with low coverage in this setting. Binomial errors are reported for empirical coverage (95%) and standard errors are reported for all other values.

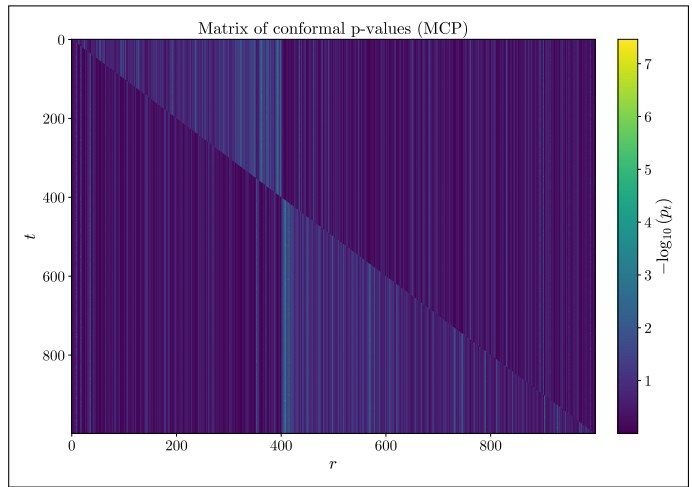

(a) matrix of conformal p-values (MCP)

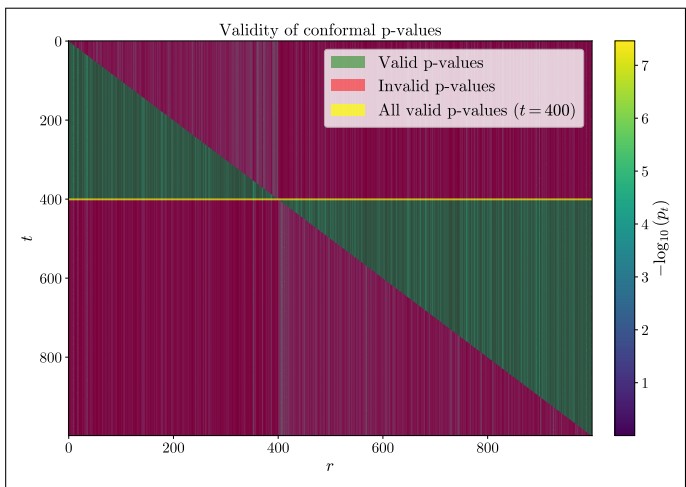

(b) validity of conformal p-values

Figure 8: (a) Matrix of conformal p-values (MCP) for a Gaussian mean change using the oracle score function. The true changepoint occurs at $\xi = 400$. (b) Validity of p-values in the matrix of conformal p-values (green p-values are valid, red p-values are invalid). Note that all of the p-values are only valid when $t = \xi = 400$, shown as the yellow row.

### 7.2 Sentiment change using large language models (LLMs)

Next, we consider a simulation with a sentiment change in a sequence of text samples, showing that our algorithm is practical for localizing changepoints in language data. We consider the Stanford Sentiment Treebank (SST-2) dataset of movie reviews labeled with a binary sentiment, introduced in Socher et al. (2013). Suppose that we would like to localize a change from a generally positive sentiment in movie reviews to a negative sentiment. In practice, this method may be applied to localize a change in customer sentiment toward a product or general approval of a political leader.

Suppose we observe $n = 1000$ reviews with a changepoint at $\xi = 400$; the pre-change class $\mathbb{P}_0$ consists of i.i.d. draws of positive reviews and the post-change class $\mathbb{P}_1$ consists of i.i.d. draws of negative reviews. For instance, here is a sample of the data before and after the changepoint:

- $t = 399$ (positive): "invigorating, surreal, and resonant with a rainbow of emotion."

- $t = 400$ (positive): "a fascinating, bombshell documentary"

- $t = 401$ (negative): "a fragment of an underdone potato"

- $t = 402$ (negative): "is a disaster, with cloying messages and irksome characters"

As described in Section 5, we can use a pretrained multi-class classifier to estimate the likelihood ratio using the data before and after $t$ (for each $t \in [n-1]$). For this simulation, we use the DistilBERT base model, fine-tuned for sentiment analysis on the uncased SST-2 dataset (Sanh et al. (2019)).

The resulting confidence set is $[368, 420]$ and is shown in Figure 9a; note that we were able to obtain such a tight confidence interval without even knowing the nature of the change that would happen. In fact, consider the more realistic (but much more difficult to localize) scenario where general sentiment changes from being 60% positive pre-change to 40% positive post-change. In this case, we are still able to form a nontrivial confidence set, shown in Figure 9b. Even though the change is extremely subtle and we impose very general modeling assumptions, we are able to obtain the 95% confidence set $[200, 475] \cup \{478, 487, 489, 493\}$ for the changepoint.

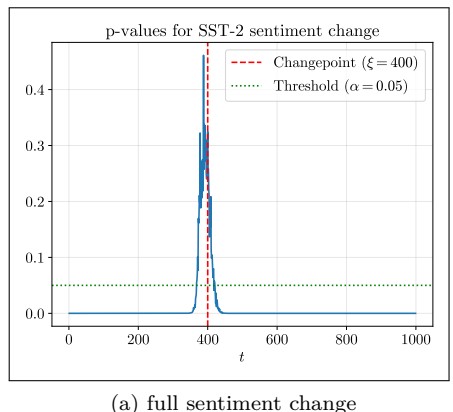
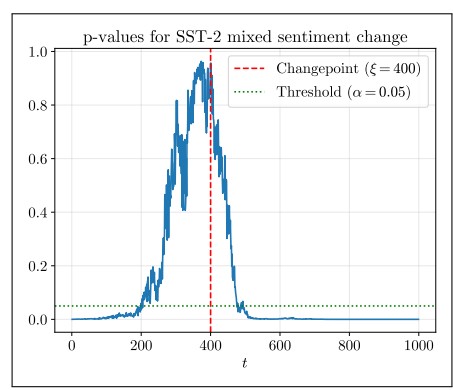

(a) full sentiment change          (b) mixed sentiment change

Figure 9: p-values for SST-2 sentiment change at $\xi = 400$ using DistilBERT trained for sentiment analysis. The dashed red line indicates the true changepoint, and the region on the horizontal axis where the p-values lie above the dotted green line ($\alpha = 0.05$) corresponds to our 95% confidence set. The point estimators are (a) $\hat{\xi} = 388$ and (b) $\hat{\xi} = 373$, both of which are close to the true $\xi = 400$.

### 7.3 Human activity change: accelerometer data

Here, we include a simulation using data from the Human Activity Recognition (HAR) dataset, introduced in Anguita et al. (2013). This dataset contains tri-axial accelerometer data collected from a smartphone

accelerometer mounted at the waist of 30 human subjects during various activities. Suppose we observe $n = 1000$ samples accelerometer with a changepoint at $\xi = 400$; the pre-change class $\mathbb{P}_0$ consists of accelerometer samples from the "walking upstairs" activity and the post-change class $\mathbb{P}_1$ consists of samples from the "standing" activity. For simplicity, we only use accelerometer data from the first axial direction, so the data we work with is real-valued. We show the dataset in Figure 10.

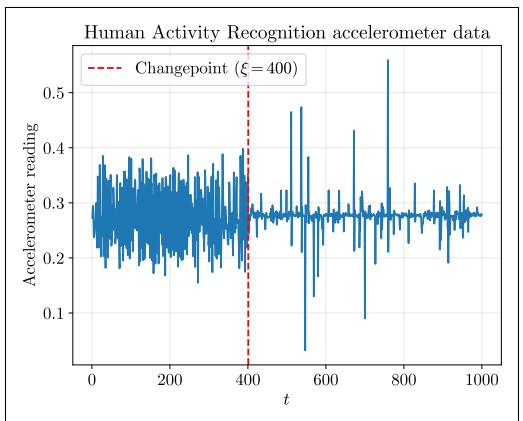

Figure 10: Human Activity Recognition (HAR) accelerometer data containing an activity change from "walking upstairs" to "standing" with a changepoint at $\xi = 400$.

Then, as described in Section 5, we can use a kernel density estimator (KDE) to estimate the likelihood ratio using the data before and after $t$ (for each $t \in [n-1]$), using a Gaussian kernel. We choose bandwidth $10^{-1}$ when $t \in \{1, n-1\}$ and bandwidth $r^{-1/5}$ otherwise, where $r$ is the number of samples used to learn each KDE; this is known as Scott's rule (due to Scott (1979)). However, due to dependence of the left and right p-values, we use the Bonferroni combining rule, plotting the resulting p-values $(p_t)_{t=1}^{n-1}$ in Figure 11.

The resulting confidence set is $[376, 444]$ and contains the changepoint, as shown in Figure 11. Note that we were able to obtain such a tight confidence interval without any assumptions about the nature of the change. As discussed in Section 2, a lot of work on prior nonparametric changepoint localization focuses on localizing a change in mean. However, for our dataset, the change in mean between the left and right halves is only $-0.003$, making it difficult to localize using this class of methods.

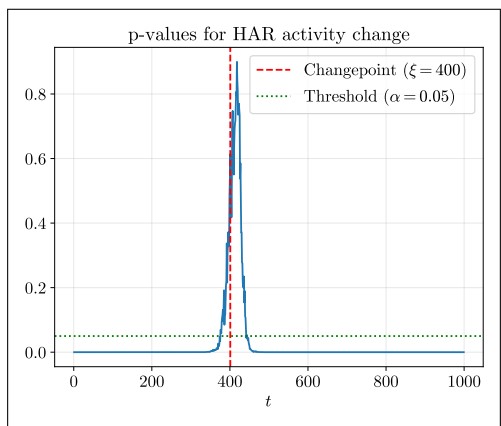

Figure 11: p-values for HAR sentiment change at $\xi = 400$ using kernel density estimators. The dashed red line indicates the true changepoint, and the region on the horizontal axis where the p-values lie above the dotted green line ($\alpha = 0.05$) corresponds to our 95% confidence set. The point estimator is $\hat{\xi} = 418$, which is close to the true changepoint at $\xi = 400$.

## 7.4 CIFAR-100 image change: bears vs. beavers

Finally, consider a simulation of a digit change from the CIFAR-100 image dataset (Krizhevsky, 2009); even when the data is high-dimensional with complex structure, our algorithm can be used to localize the changepoint. This dataset contains $32 \times 32 \times 3$ RGB images from 100 classes, making the localization problem challenging. Suppose that we observe data of length $n = 800$ and the changepoint is $\xi = 300$; the pre-change class $\mathbb{P}_0$ consists of i.i.d. draws from the set of bear images and the post-change class $\mathbb{P}_1$ consists of i.i.d. draws from the set of beaver images (Figure 12). Note that bears and beavers are visually difficult to tell apart, so localization of the changepoint is a difficult task.

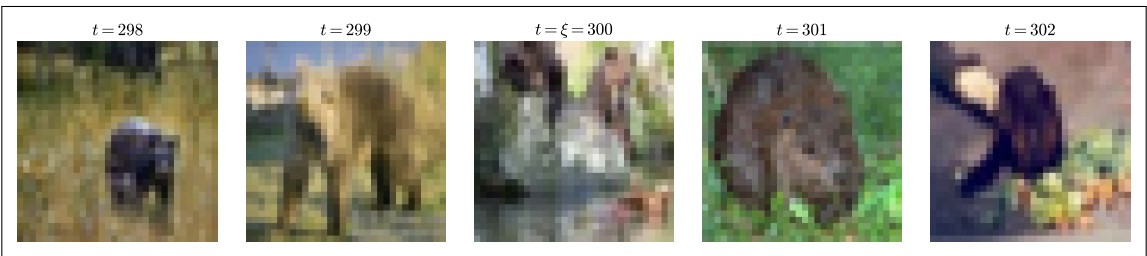

Figure 12: Partial sample of CIFAR-100 change from bear to beaver images with a changepoint at $\xi = 300$.

As described in Section 5, we can use a pretrained multi-class classifier to estimate the likelihood ratio using the data before and after $t$ (for each $t \in [n-1]$). We use the resnet18_cifar100 model from Hugging Face, trained by Eduardo Dadalto, which achieves $79.26\%$ accuracy on the test dataset.

We can then use the MCP algorithm and Algorithm 2 to produce a confidence interval. In this case, the left and right p-values are independent, since the left score function $s_t^{(0)}$ only depends on data to the left of $t$ and the right score function $s_t^{(1)}$ only depends on data to the right of $t$. Therefore, we use the minimum method to combine the p-values, plotting the resulting p-values $(p_t)_{t=1}^{n-1}$ in Figure 13; the resulting confidence set is $[263, 316]$. For a similar experiment regarding a digit change from the MNIST dataset (Deng (2012)), see Appendix A.

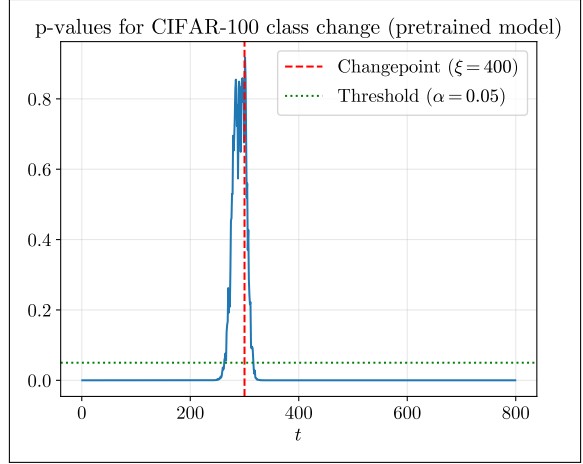

Figure 13: p-values for CIFAR-100 change from bears to beavers at $\xi = 300$ using a pre-trained digit classifier. The dashed red line indicates the true changepoint, and the region on the horizontal axis where the p-values lie above the dotted green line ($\alpha = 0.05$) corresponds to our 95% confidence set. The point estimator is $\hat{\xi} = 301$, which is close to the true $\xi = 300$.

# 8   Conclusion

We introduced the MCP algorithm, a novel method for constructing confidence sets for the changepoint, assuming that the pre-change and post-change distributions are each exchangeable. We demonstrated that the MCP algorithm can be used to construct nontrivial confidence sets for the changepoint in a variety of settings and enjoys a finite-sample coverage guarantee. Furthermore, we demonstrated through simulation and experiments that the algorithm has good empirical coverage and width. We also discussed how the parameters of the MCP algorithm, including the score function, should be chosen in practice to produce tight confidence intervals.

### Acknowledgments

We thank Carlos Padilla for early discussions, Jing Lei for discussions relating to the conformal Neyman-Pearson lemma, Swapnaneel Bhattacharyya for noticing an error in an early preprint, Alessandro Rinaldo for several references to prior work in changepoint localization, and Rohan Hore for providing an implementation of the KCPD algorithm.

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

# A MNIST digit change

In this section, we consider a simulation of a digit change from the MNIST handwritten digit dataset (Deng, 2012). Suppose that we observe data of length $n = 1000$ and the changepoint is $\xi = 400$; the pre-change class $\mathbb{P}_0$ consists of i.i.d. draws from the set of handwritten "3" digits and the post-change class $\mathbb{P}_1$ consists of i.i.d. draws from the set of handwritten "7" digits (Figure 14).

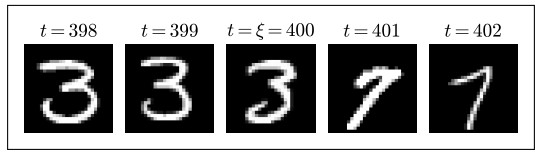

Figure 14: Partial sample of MNIST digit change from "3" to "7" with a changepoint at $\xi = 400$.

As described in Section 5, we can use a pretrained multi-class classifier to estimate the likelihood ratio using the data before and after $t$ (for each $t \in [n-1]$). In this case, we use a simple convolutional neural network; see Appendix H for details about how the network was trained.

We can then use the MCP algorithm and Algorithm 2 to produce a confidence interval. In this case, the left and right p-values are independent, since the left score function $s_t^{(0)}$ only depends on data to the left of $t$ and the right score function $s_t^{(1)}$ only depends on data to the right of $t$. Therefore, we use the minimum method to combine the p-values, plotting the resulting p-values $(p_t)_{t=1}^{n-1}$ in Figure 15.

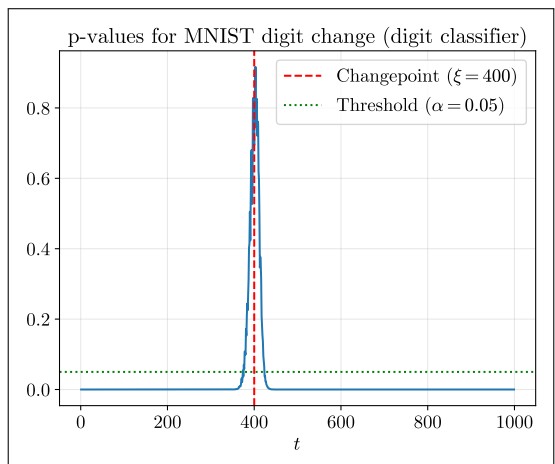

Figure 15: p-values for MNIST digit change at $\xi = 400$ using a pre-trained digit classifier. The dashed red line indicates the true changepoint, and the region on the horizontal axis where the p-values lie above the dotted green line ($\alpha = 0.05$) corresponds to our 95% confidence set. The point estimator is $\hat{\xi} = 403$, which is close to the true $\xi = 400$.

In particular, the resulting confidence set is $[374, 423]$, which is completely nontrivial and obtained only using a classifier learned on the entire MNIST dataset. Note also that our confidence sets can be significantly tightened if we are given access to certified pre-change or post-change samples; see Appendix B for related experiments.

# B Using certified data to tighten confidence sets

Consider the MNIST digit change experiment of Appendix A but suppose we had additional certified pre-change and post-change samples, which we add to the left and right samples respectively. In this case, we are able to obtain better results.

When we only have additional pre-change samples, we get the confidence interval $[384, 407]$, so the additional pre-change samples allow us to tighten our confidence interval. We show this confidence set in Figure 16a.

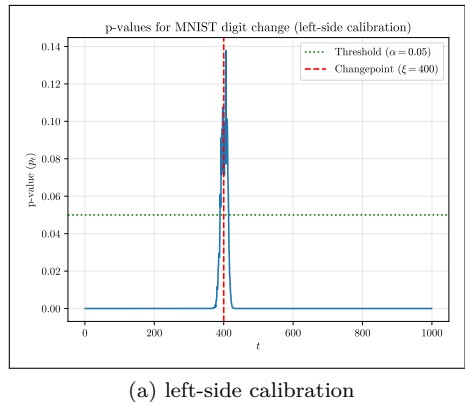
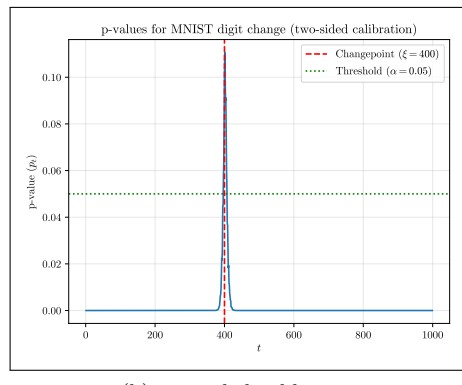

(a) left-side calibration         (b) two-sided calibration

Figure 16: p-values for an MNIST digit change at $\xi = 400$ using 100 certified calibration samples (a) on the left side only and (b) on each side. The dashed red line indicates the true changepoint, and the region on the horizontal axis where the p-values lie above the dotted green line ($\alpha = 0.05$) corresponds to our 95% confidence set. The point estimators are (a) $\hat{\xi} = 393$ and (b) $\hat{\xi} = 405$, both of which are close to the true $\xi = 400$.

If we assume we have access to 100 pre-change and post-change samples (i.e., we can learn the nature of the change that will happen), we similarly obtain the tighter confidence interval $[390, 412]$, shown in Figure 16b.

## C  Comparison of combining functions

In this section, we compare empirically between different combining functions. We consider a dataset of size $n = 100$ containing a change from $\mathcal{N}(-1, 1)$ to $\mathcal{N}(1, 1)$ at $\xi = 40$. We apply the MCP algorithm to this dataset using the minimum, Fisher, and Bonferroni combining methods outlined in Section 4.2 with the oracle and KDE score functions used in Section 7.1, in order to construct a 95% confidence set. Since the KDE score function is adaptive and thereby induces dependence between the left and right p-values, our theoretical results are technically only valid if one uses the Bonferroni combining method. However, we observe that in practice, the choice of combining method has very little effect on the resulting coverage and width of our confidence sets. We display the results in Table 4.

| Method | Score function | Avg. width | Empirical coverage | Bias | Mean abs. dev. |
|---|---|---|---|---|---|
| Minimum | Oracle | $30.8_{\text{se: }10.5}$ | $0.95_{\pm 0.03}$ | $-0.5_{\text{se: }6.8}$ | $4.9_{\text{se: }4.7}$ |
| | KDE | $33.1_{\text{se: }12.3}$ | $0.95_{\pm 0.03}$ | $1.4_{\text{se: }14.8}$ | $7.7_{\text{se: }12.7}$ |
| Fisher | Oracle | $31.9_{\text{se: }11.3}$ | $0.95_{\pm 0.03}$ | $-0.5_{\text{se: }6.1}$ | $4.5_{\text{se: }4.2}$ |
| | KDE | $32.9_{\text{se: }12.7}$ | $0.95_{\pm 0.03}$ | $0.1_{\text{se: }13.9}$ | $7.5_{\text{se: }11.7}$ |
| Bonferroni | Oracle | $31.4_{\text{se: }10.7}$ | $0.95_{\pm 0.03}$ | $-2.8_{\text{se: }6.8}$ | $5.7_{\text{se: }4.6}$ |
| | KDE | $33.6_{\text{se: }13.0}$ | $0.95_{\pm 0.03}$ | $1.0_{\text{se: }14.4}$ | $7.7_{\text{se: }12.2}$ |

Table 4: Average width and coverage of confidence sets for Gaussian mean change, as well as the bias ($\mathbb{E}[\hat{\xi} - \xi]$) and mean absolute deviation ($\mathbb{E}[|\hat{\xi} - \xi|]$) of the point estimator over 1000 trials. The length of the data is $n = 100$ and we observe a change from $\mathcal{N}(-1, 1)$ to $\mathcal{N}(1, 1)$ at $\xi = 40$. We run the MCP algorithm using the oracle and KDE score functions (Section 4.2) to construct a 95% confidence set for the changepoint. Note that the choice of combining function neither significantly affects the width and coverage of the resulting confidence sets, nor the bias and mean absolute deviation of our point estimator. Binomial errors are reported for empirical coverage (95%) and standard errors are reported for all other values.

## D  Extensions of the MCP algorithm

Here, we describe several alternatives to the MCP algorithm described in Section 3 and Section 4.

### D.1 Alternative tests

First, we detail several alternatives to the empirical test detailed in Section 4.1 for constructing the left and right p-values.

**Asymptotic KS test**  We can use the asymptotic distribution of $W_t$ to construct confidence sets for the changepoint. This method is valid when $t$ and $n-t$ are both large. Recall that $\varphi_t = B_t - t B_1$ (where $(B_t)_{t \geq 0}$ is a standard Brownian motion on $\mathbb{R}$) is the Brownian bridge on $\mathbb{R}$. Define $F_\varphi(z) := \mathbb{P}\left(\sup_{t \in [0,1]} |\varphi_t| \leq z\right)$, where $(\varphi_t)_{t \geq 0}$ is the Brownian bridge. Then, by a central limit theorem due to Donsker (1951), we can compute the asymptotically valid p-values

$$p_t^{\text{left}} = 1 - F_\varphi(W_t^{(0)}), \qquad p_t^{\text{right}} = 1 - F_\varphi(W_t^{(1)}).$$

Note that in general, one can use the approximation $p_t^{\text{left}} \approx 2 \exp(-2\,(W_t^{(0)})^2)$, which is obtained by truncating the series expansion of the limiting distribution (Kolmogorov (1933)). In practice, we can use the empirical test to test $\mathcal{H}_{0t}$ when $t$ and $n-t$ are small, and use the asymptotic test when $t$ and $n-t$ are large. Next, we will discuss how to combine the left and right p-values from Section 4.1 into a single p-value for $\mathcal{H}_{0t}$.

If the asymptotic test is used, we have the following asymptotic coverage guarantee for the MCP algorithm, analogous to the one given in Section 4.1.

**Theorem D.1** (Asymptotic coverage guarantee (asymptotic KS test)). *Consider the setting of Theorem 4.1 with a changepoint $\xi = \lfloor \gamma n \rfloor$ for some $\gamma \in (0,1)$, but where $\mathcal{C}_{1-\alpha}$ is instead constructed using the asymptotic test described in Section 4.1. Then, the coverage of $\mathcal{C}_{1-\alpha}$ is asymptotically at least $1 - \alpha$:*

$$\liminf_{n \to \infty} \mathbb{P}\left(\xi \in \mathcal{C}_{1-\alpha}\right) \geq 1 - \alpha.$$

**Permutation test**  Instead of using the empirical test or asymptotic KS test, we can use permutation tests to construct confidence sets for the changepoint by using the distribution of $W_t^{(0)}$ and $W_t^{(1)}$ under permutations of the data before and after $t$ respectively. Under the null hypothesis $\mathcal{H}_{0t}$, the data before and after $t$ is exchangeable, so the distribution of $(W_t^{(0)}, W_t^{(1)})$ under permutations of the data before and after $t$ should remain the same. This method is valid for all values of $t$, and we can use the following algorithm to construct confidence sets for the changepoint using permutation tests.

---

**Algorithm 3:** Permutation test

**Input:** $(W_t^{(0)}, W_t^{(1)})$ (discrepancy scores), $B \in \mathbb{N}$ (user-chosen sample size)
**Output:** $(p_t^{\text{left}}, p_t^{\text{right}})$ (a pair of p-values for the left and right data)

1  **for** $b \in [B]$ **do**
2  $\quad$ $(X_t^{(b)})_{t=1}^n \sim \text{Unif}([0,1]^n)$
3  $\quad$ Compute $(W_t^{(0,b)})_{t=1}^{n-1}$ and $(W_t^{(1,b)})_{t=1}^{n-1}$ using Line 1 to Line 15 of Algorithm 1 on permuted data $(X_t^{(b)})_{t=1}^n$
4  **end**
5  Draw $\theta_0, \theta_1 \sim \text{Unif}(0,1)$
6  $p_t^{\text{left}} \leftarrow \frac{1}{B+1}\left(\theta_0 + \sum_{b=1}^B (\mathbf{1}_{W_t^{(0,b)} > W_t^{(0)}} + \theta_0\, \mathbf{1}_{W_t^{(0,b)} = W_t^{(0)}})\right)$
7  $p_t^{\text{right}} \leftarrow \frac{1}{B+1}\left(\theta_1 + \sum_{b=1}^B (\mathbf{1}_{W_t^{(1,b)} > W_t^{(1)}} + \theta_1\, \mathbf{1}_{W_t^{(1,b)} = W_t^{(1)}})\right)$
8  **return** $(p_t^{\text{left}}, p_t^{\text{right}})$

---

In practice, we have observed that permutation testing performs very similarly to the empirical test, but it is much slower to run. Therefore, we recommend using the empirical test and asymptotic test over the permutation test. In addition, the empirical test can be run once to obtain empirical distributions for the discrepancy scores, and those distributions can be reused across datasets; this is not possible with the permutation test, which must be run on each dataset.

### D.2 Alternatives to the Kolmogorov-Smirnov statistic

Instead of using the Kolmogorov-Smirnov statistic to measure the discrepancy between the normalized ranks and the uniform distribution, we can use other discrepancy measures such as the Cramér-von Mises statistic, the Anderson-Darling statistic, the Kuiper statistic, or the Wasserstein $p$-distance. These statistics all give rise to valid hypothesis tests of $\mathcal{H}_{0t}$, and we can use them to construct confidence sets for the changepoint using the same methods described in Section 4 using the MCP algorithm. However, in our experiments, we have not observed a significant difference in performance between the Kolmogorov-Smirnov statistic and these other statistics.

## E Confidence sets with no changepoint

When there is no changepoint, Algorithm 1 tends to produce very wide confidence sets, but which typically include $n$ (meaning that the practitioner can correctly conclude that there is a possibility that there is no changepoint in the dataset). We summarize these findings in Table 5.

| $\alpha$ | Avg. width | Empirical coverage |
|---|---|---|
| 0.05 | $476.0_{\text{se: }53.9}$ | $0.95_{\pm 0.03}$ |
| 0.5 | $250.4_{\text{se: }136.5}$ | $0.51_{\pm 0.03}$ |

Table 5: Average width and coverage of confidence sets for Gaussian mean change when there is no changepoint. The length of the data is $n = 500$ and all data is from $\mathcal{N}(-1, 1)$. We run the MCP algorithm using the oracle score function (Section 4.2) assuming that there is a change from $\mathcal{N}(-1, 1)$ to $\mathcal{N}(1, 1)$ in order to construct a $1 - \alpha$ confidence set for the changepoint. We note that the resulting confidence sets are wide, but typically have the correct coverage. Binomial errors are reported for empirical coverage and standard errors are reported for the width.

## F Proofs of main results

In this section, we present proofs of our main results. For an alternative proof technique, see Vovk et al. (2003); Angelopoulos et al. (2024). We begin with a useful lemma (for more on this lemma, see Brockwell (2007)).

**Lemma F.1** (Randomized probability integral transform)**.** *Suppose $X$ is a real-valued random variable with cdf $F_X$ and let $V \sim \text{Unif}(0, 1)$ be drawn independently of $X$. If we define*

$$U = \lim_{y \uparrow X} F_X(y) + V\left(F_X(X) - \lim_{y \uparrow X} F_X(y)\right),$$

*then $U \sim \text{Unif}(0, 1)$. Here, $U$ is called the randomized probability integral transform of $X$.*

*Proof of Lemma F.1.* Let $F_X^{-1}(x) = \inf\{z \in \mathbb{R} : F_X(z) \geq x\}$ denote the quantile transformation of $X$. We then fix $u \in [0, 1]$ and decompose

$$\mathbb{P}(U \leq u) = \mathbb{P}(U \leq u, X < F_X^{-1}(u)) + \mathbb{P}(U \leq u, X = F_X^{-1}(u)) + \mathbb{P}(U \leq u, X > F_X^{-1}(u)). \tag{1}$$

We proceed by estimating these three quantities. Note that for any $x < F_X^{-1}(u)$, we have

$$\lim_{y \uparrow x} F_X(y) \leq F_X(x) < u.$$

This means that if $X < F^{-1}(u)$ we have

$$
\begin{aligned}
U &= \lim_{y \uparrow X} F_X(y) + V \left( F_X(X) - \lim_{y \uparrow X} F_X(y) \right) \\
&< \lim_{y \uparrow X} F_X(y) + V \left( u - \lim_{y \uparrow X} F_X(y) \right) \\
&= (1 - V) \lim_{y \uparrow X} F_X(y) + V u \\
&< (1 - V) u + V u \\
&= u.
\end{aligned}
$$

Therefore, the first term in Equation (1) is

$$
\mathbb{P}(U \leq u, \, X < F_X^{-1}(u)) = \mathbb{P}(X < F_X^{-1}(u)) = \lim_{y \uparrow F_X^{-1}(u)} F_X(y).
$$

Next, suppose that $X = F_X^{-1}(u)$. If $F_X$ is continuous at $X$, then it is obvious that $U = u$. If not, then we have

$$
U = \lim_{y \uparrow X} F_X(y) + V \left( F_X(X) - \lim_{y \uparrow X} F_X(y) \right) \leq u \iff V \leq \frac{u - \lim_{y \uparrow X} F_X(y)}{F_X(X) - \lim_{y \uparrow X} F_X(y)}.
$$

Hence, we find that

$$
\begin{aligned}
&\mathbb{P}\left( U \leq u, \, X = F_X^{-1}(u) \right) \\
&= \mathbb{P}(X = F_X^{-1}(u)) \, \mathbb{P}\left( V \leq \frac{u - \lim_{y \uparrow X} F_X(y)}{F_X(X) - \lim_{y \uparrow X} F_X(y)} \,\bigg|\, X = F_X^{-1}(u) \right) \\
&= \left( F_X(F_X^{-1}(u)) - \lim_{y \uparrow F_X^{-1}(u)} F_X(y) \right) \mathbb{P}\left( V \leq \frac{u - \lim_{y \uparrow F_X^{-1}(u)} F_X(y)}{F_X(F_X^{-1}(u)) - \lim_{y \uparrow F_X^{-1}(u)} F_X(y)} \right).
\end{aligned}
$$

Note that $\lim_{y \uparrow F_X^{-1}(u)} F_X(y) \leq u \leq F_X(F_X^{-1}(u))$ by definition, so since $V \sim \mathrm{Unif}(0, 1)$ we obtain

$$
\begin{aligned}
&\left( F_X(F_X^{-1}(u)) - \lim_{y \uparrow F_X^{-1}(u)} F_X(y) \right) \mathbb{P}\left( V \leq \frac{u - \lim_{y \uparrow F_X^{-1}(u)} F_X(y)}{F_X(F_X^{-1}(u)) - \lim_{y \uparrow F_X^{-1}(u)} F_X(y)} \right) \\
&= \left( F_X(F_X^{-1}(u)) - \lim_{y \uparrow F_X^{-1}(u)} F_X(y) \right) \left( \frac{u - \lim_{y \uparrow F_X^{-1}(u)} F_X(y)}{F_X(F_X^{-1}(u)) - \lim_{y \uparrow F_X^{-1}(u)} F_X(y)} \right) \\
&= u - \lim_{y \uparrow F_X^{-1}(u)} F_X(y).
\end{aligned}
$$

Whether $F_X$ is continuous at $X$ or not, the second term in Equation (1) is given by

$$
\mathbb{P}\left( U \leq u, \, X = F_X^{-1}(u) \right) = u - \lim_{y \uparrow F_X^{-1}(u)} F_X(y).
$$

Finally, consider the case $X > F_X^{-1}(u)$ in Equation (1). By monotonicity of the cdf, we have

$$
\lim_{y \uparrow X} F_X(y) \geq F_X(F_X^{-1}(u)).
$$

By definition, $F_X(F_X^{-1}(u)) \geq u$, so we see that

$$
U = \lim_{y \uparrow X} F_X(y) + V \left( F_X(X) - \lim_{y \uparrow X} F_X(y) \right) \geq \lim_{y \uparrow X} F_X(y) \geq u.
$$

Thus, $\{U \le u, X > F_X^{-1}(u)\}$ can only happen if $U = u$. Except for the zero-probability event where $V = 0$, this also forces $X \in S := \{x \in \mathbb{R} : \lim_{y \uparrow x} F_X(y) = F_X(x) = u\}$ and because $F_X$ is constant on $S$, we find that

$$\mathbb{P}(U \le u, X > F_X^{-1}(u)) \le \mathbb{P}(X \in S) = 0.$$

We deduce the result from summing the three terms in Equation (1). $\qquad\square$

Using Lemma F.1, we can prove coverage guarantees for our confidence sets.

*Proof of Theorem 4.1.* First, we demonstrate that $p_r^{(t)}$ is always a true p-value under the null $\mathcal{H}_{0t}$ (conditional on the bag $[\![X_1, \ldots, X_r]\!]$), for all $r \in [t]$. We may assume without loss of generality that $r \le t$, since the case $t < r \le n$ follows by a symmetric argument. Since $X_1, \ldots, X_t$ are exchangeable under $\mathcal{H}_{0t}$, for any permutation $\pi : [r] \to [r]$ we have

$$
\left( \kappa_{r, \pi(1)}^{(t)}, \ldots, \kappa_{r, \pi(r)}^{(t)} \right)
$$
$$
= \left( s_t^{(0)}(X_{\pi(1)}; [\![X_1, \ldots, X_r]\!], (X_{t+1}, \ldots, X_n)), \ldots, s_t^{(0)}(X_{\pi(r)}; [\![X_1, \ldots, X_r]\!], (X_{t+1}, \ldots, X_n)) \right)
$$
$$
= \pi(\kappa_{r1}^{(t)}, \ldots, \kappa_{rr}^{(t)}).
$$

Hence, the scores $\kappa_{r1}^{(t)}, \ldots, \kappa_{rr}^{(t)}$ are exchangeable under $\mathcal{H}_{0t}$. Now, if $\tilde{U} \sim \mathrm{Unif}([r])$, then the cdf of $X_{\tilde{U}}$ conditional on the bag of observations $[\![X_1, \ldots, X_r]\!]$ is

$$F_{X_{\tilde{U}}}(x) = \frac{1}{r} \sum_{j=1}^r \mathbf{1}_{\kappa_{rj}^{(t)} \le x}.$$

Conditional on the bag of observations, $\kappa_{rr}^{(t)}$ is distributed like $X_{\tilde{U}}$, so by the randomized probability integral transform and the fact that $1 - \theta_r^{(t)} \sim \mathrm{Unif}(0,1)$, we find that (conditional on $[\![X_1, \ldots, X_r]\!]$)

$$U = \frac{1}{r} \sum_{j=1}^r \mathbf{1}_{\kappa_{rj}^{(t)} < \kappa_{rr}^{(t)}} + \frac{1 - \theta_r^{(t)}}{r} \sum_{j=1}^r \mathbf{1}_{\kappa_{rj}^{(t)} = \kappa_{rr}^{(t)}} \sim \mathrm{Unif}(0,1).$$

In particular, this means that (conditional on $[\![X_1, \ldots, X_r]\!]$)

$$
1 - U = \frac{1}{r} \sum_{j=1}^r \mathbf{1}_{\kappa_{rj}^{(t)} \ge \kappa_{rr}^{(t)}} - \frac{1 - \theta_r^{(t)}}{r} \sum_{j=1}^r \mathbf{1}_{\kappa_{rj}^{(t)} = \kappa_{rr}^{(t)}}
$$
$$
= \frac{1}{r} \sum_{j=1}^r \mathbf{1}_{\kappa_{rj}^{(t)} > \kappa_{rr}^{(t)}} + \frac{\theta_r^{(t)}}{r} \sum_{j=1}^r \mathbf{1}_{\kappa_{rj}^{(t)} = \kappa_{rr}^{(t)}}
$$
$$
= p_r^{(t)} \sim \mathrm{Unif}(0,1).
$$

Integrating over $[\![X_1, \ldots, X_r]\!]$ shows that $p_r^{(t)}$ is an unconditional p-value for $\mathcal{H}_{0t}$ such that $\{p_r^{(t)}\}_{r=1}^t$ are independent (Vovk et al., 2003, Proposition 3.1). Of course, this means that under the null hypothesis, the discrepancy scores $((W_t^{(i)})_{i=0}^1)_{t=1}^{n-1}$ are distributed as if the data was all i.i.d. from $\mathrm{Unif}(0,1)$. Another application of the randomized probability integral transform (Lemma F.1) gives that $p_t^{\mathrm{left}}$ and $p_t^{\mathrm{right}}$ are exactly uniformly distributed, and any of the methods in Section 4.2 will yield a valid confidence interval by the Neyman construction. $\qquad\square$

*Proof of Theorem D.1.* The proof of this fact is completely analogous to the proof of Theorem 4.1, except that the test used is only asymptotically valid as $t \to \infty$ and $n-t \to \infty$ simultaneously. Letting $\varphi_t = B_t - tB_1$ (where $(B_t)_{t \ge 0}$ denotes a standard Brownian motion on $\mathbb{R}$), the validity of the test follows from the following theorem of Donsker (1951):

$$\sqrt{t} \left( \sup_{z \in \mathbb{R}} |\hat{F}_0(z) - u(z)| \right) = \sqrt{t}\, \mathrm{KS}(\hat{F}_0, u) \xrightarrow{d} \sup_{t \in [0,1]} |\varphi_t|. \qquad\square$$

*Proof of Theorem 5.1.* Let $F_Y$ denote the cdf of the random variable $Y$ and let $F_Y^{-1}(y) = \inf\{z \in \mathbb{R} : F_Y(z) \geq y\}$ denote the quantile transformation. Now, it's clear from definitions that

$$\mathbb{E}[T_n[s]] = \frac{1}{n} \sum_{i=1}^n \left( \mathbb{P}(s(X_i) < s(X_{n+1})) + \frac{1}{2} \mathbb{P}(s(X_i) = s(X_{n+1})) \right).$$

The second term does not depend on the choice of $s$, so it suffices to optimize over the first term. By the randomized probability integral transform (Lemma F.1), we obtain

$$\mathbb{P}(s(X_1) < s(X_{n+1})) + \frac{1}{2} \mathbb{P}(s(X_1) = s(X_{n+1}))$$

$$= \mathbb{P}\left( \lim_{y\uparrow s(X_1)} F_{s(X_{n+1})}(y) + \theta_1 \left( F_{s(X_{n+1})}(s(X_1)) - \lim_{y\uparrow s(X_1)} F_{s(X_{n+1})}(y) \right) < U \right),$$

for an independent uniform $U \sim \mathrm{Unif}(0,1)$. Define

$$F_Y^*(s(X_1)) := \lim_{y\uparrow s(X_1)} F_Y(y) + \theta_1 \left( F_Y(s(X_1)) - \lim_{y\uparrow s(X_1)} F_Y(y) \right).$$

Let $\mathbb{P}(F_{s(X_{n+1})}^*(s(X_1)) < u \,|\, U = u)$ be a regular conditional probability, which exists and is unique for Lebesgue-almost every $u \in (0,1)$ by the disintegration of measure. Hence, the above expression equals

$$\mathbb{E}_U[\mathbb{P}(F_{s(X_{n+1})}^*(s(X_1)) < u \,|\, U = u)] = \int_0^1 \mathbb{P}\left( F_{s(X_{n+1})}^*(s(X_1)) < u \right) du.$$

Fix $u \in (0,1)$ and suppose we want to test $\tilde{\mathcal{H}}_0 : X_1 \sim Q$ against $\tilde{\mathcal{H}}_1 : X_1 \sim R$, rejecting $\tilde{\mathcal{H}}_0$ whenever $F_{s(X_{n+1})}^*(s(X_1)) \leq u$. Under $\tilde{\mathcal{H}}_0$, the cdf of $s(X_1)$ is exactly $F_{s(X_{n+1})}$, so the randomized probability integral transform (Lemma F.1) gives that the Type I error of our test is $u$:

$$\mathbb{P}_{\tilde{\mathcal{H}}_0}\left( F_{s(X_{n+1})}^*(s(X_1)) < u \right) = \mathbb{P}_{\tilde{\mathcal{H}}_0}\left( F_{s(X_1)}^*(s(X_1)) < u \right) = u.$$

The power of our test is $\mathbb{P}_{\tilde{\mathcal{H}}_1}\left( F_{s(X_{n+1})}^*(s(X_1)) < u \right)$. By the generalized Neyman-Pearson lemma (see, for example, Theorem 3.2.1 and Section 5.9 of Lehmann & Romano (2005)), the power is maximized by the choice $s = s^*$. Integrating the bound from 0 to 1 gives the desired result.

The usual Neyman-Pearson lemma (Lehmann & Romano, 2005, Theorem 3.2.1 (ii)) states that any test $\phi$ of exact size $\alpha$ of the form $\phi(z) = \mathbf{1}_{s^*(z) \geq t_\alpha}$ is most powerful for testing $\mathcal{H}_0$ against $\mathcal{H}_1$ at level $\alpha$. Hence, we deduce that the choice $s^*$ maximizes the power of any level $\alpha$ test of $\mathcal{H}_0$ against $\mathcal{H}_1$. Furthermore, we know that $\phi_\alpha[s]$ is an exact test at size $\alpha$ ($\mathbb{E}[\phi_\alpha[s]] = \alpha$ when $Q = R$) which is independent of the choice of score $s$ (Angelopoulos et al., 2024, Lemma 9.3). Putting the pieces together, the last statement of the theorem follows. $\square$

## G   The log-optimal e-variable uses the likelihood ratio score

While the reasoning in Theorem 5.1 yields that conformal *p-values* are optimized by the likelihood ratio score, it turns out to also be true that conformal *e-values* are also optimized by the same score. Indeed, in order to test exchangeability of $X_1, \ldots, X_{n+1}$ against the alternative that their joint distribution is given by $R^n \times Q$, one can show (Ramdas & Wang, 2025, Section 6.7.7) that the log-optimal e-variable is given by

$$\frac{q(X_{n+1}) \prod_{i=1}^n r(X_i)}{\frac{1}{n+1} \sum_{i=1}^{n+1} q(X_i) \prod_{j \neq i} r(X_j)}.$$

Dividing throughout by $\prod_{i=1}^{n+1} r(X_i)$, the above e-variable becomes

$$\frac{q(X_{n+1})/r(X_{n+1})}{\frac{1}{n+1} \sum_{i=1}^{n+1} q(X_i)/r(X_i)}.$$

In fact, every e-variable for testing exchangeability must be of the form $\frac{s(\mathbf{X})}{\sum_\pi s(\mathbf{X}_\pi)}$, where the sum is taken over all $(n+1)!$ permutations $\pi$ and $\mathbf{X} = (X_1, \ldots, X_n)$ and $\mathbf{X}_\pi = (X_{\pi(1)}, \ldots, X_{\pi(n)})$ for a positive score function $s$; thus, we see once again the the log-optimal e-variable is obtained using the likelihood ratio score

$$s(\mathbf{X}) = q(X_{n+1})/r(X_{n+1}).$$

## H   Convolutional neural network architecture

We trained a convolutional neural network from scratch on the MNIST handwritten digit dataset, using the following architecture:

- Convolutional layer (1 input channel, 32 output channels, $3 \times 3$ kernel, stride 1)

- ReLU activation

- Convolutional layer (32 input channels, 64 output channels, $3 \times 3$ kernel, stride 1)

- ReLU activation

- Max pooling layer ($2 \times 2$)

- Dropout ($p = 0.25$)

- Flattening layer

- Linear layer (output size 128)

- ReLU activation

- Dropout ($p = 0.5$)

- Linear layer (output size 10)

We train for one epoch using the Adam optimizer and cross-entropy loss, and achieve 98.63% accuracy on the test dataset. The entire training process takes approximately one minute (using the CPU) on a MacBook Pro with an M1 Pro processor.

## I   Additional methods for choosing the score function

In this section, we detail several additional methods by which the score function can be chosen in the MCP algorithm. Suppose that $\mathbb{P}_0 = \otimes_{t=1}^{\xi} \mathbb{P}_0^*$ and $\mathbb{P}_1 = \otimes_{t=\xi+1}^{n} \mathbb{P}_1^*$, where $\mathbb{P}_0^*$ and $\mathbb{P}_1^*$ have densities $f_0$ and $f_1$ respectively; here are several ways to choose the score function:

- **Likelihood ratio**: If the pre-change and post-change distributions are known, the likelihood ratio is a powerful score function (motivated by Theorem 5.1):

$$s_{rt}^{(0)}(x_r; [\![x_1, \ldots, x_r]\!], (x_{t+1}, \ldots, x_n)) = \frac{f_1(x_r)}{f_0(x_r)},$$
$$s_{rt}^{(1)}(x_r; [\![x_{r+1}, \ldots, x_n]\!], (x_1, \ldots, x_t)) = \frac{f_0(x_r)}{f_1(x_r)}.$$

- **Density estimators**: If the pre-change and post-change distributions are unknown, one can estimate the likelihood ratio using density estimators on the left and right-hand sides respectively. For instance, suppose one has trained a density estimator $\hat{f}_0$ using the data $(x_1, \ldots, x_t)$ and another density estimator $\hat{f}_1$ using the data $(x_{t+1}, \ldots, x_n)$. Furthermore, suppose one has trained a density estimator $\hat{f}_0^{\text{exch}}$ using the data $[\![x_1, \ldots, x_r]\!]$ and another density estimator $\hat{f}_1^{\text{exch}}$ using the data

$[\![x_{r+1}, \ldots, x_n]\!]$. For instance, one could use a weighted ERM to learn $\hat{f}_0$ which places more weight on earlier samples (which are more likely to come from the pre-change distribution $\mathbb{P}_0$. Then, we define the score functions as:

$$s_{rt}^{(0)}(x_r; [\![x_1, \ldots, x_r]\!], (x_{t+1}, \ldots, x_n)) = \frac{\hat{f}_1(x_r)}{\hat{f}_0^{\mathrm{exch}}(x_r)},$$

$$s_{rt}^{(1)}(x_r; [\![x_{r+1}, \ldots, x_n]\!], (x_1, \ldots, x_t)) = \frac{\hat{f}_0(x_r)}{\hat{f}_1^{\mathrm{exch}}(x_r)}.$$

For example, one could use a kernel density estimator or a neural network to learn these.

- **Classifier-based likelihood ratio**: One can use a classifier to estimate the likelihood ratio, which allows us to use our method even when the likelihood is intractable. Suppose we have access to a pre-trained classifier $\hat{g}_1 : \mathcal{X} \to [0, 1]$ which outputs an estimated probability that $x \in \mathcal{X}$ was drawn from $\mathbb{P}_1$ (instead of $\mathbb{P}_0$). Then, we can use this classifier to estimate the likelihood ratio:

$$s_{rt}^{(0)}(x_r; [\![x_1, \ldots, x_r]\!], (x_{t+1}, \ldots, x_n)) = \frac{\hat{g}_1(x_r)}{1 - \hat{g}_1(x_r)},$$

$$s_{rt}^{(1)}(x_r; [\![x_{r+1}, \ldots, x_n]\!], (x_1, \ldots, x_t)) = \frac{1 - \hat{g}_1(x_r)}{\hat{g}_1(x_r)}.$$

