# OpenReview forum: "Offline changepoint localization using a matrix of conformal p-values"
_TMLR — Accepted by TMLR_

### Review · Reviewer_MCyp · 2025-12-01

**Summary Of Contributions:**

1. To my knowledge this is the first method that gives finite sample, distribution free confidence sets for the location of a single changepoint under only exchangeability of each side.

2. The matrix of conformal p values is an elegant idea that bridges conformal prediction, two sample testing and changepoint localization.

3. They prove a coverage guarantee for the resulting confidence sets and establishes a “conformal Neyman–Pearson” lemma that shows likelihood ratio style scores are optimal within the proposed conformal framework.

4. They demonstrates the method on synthetic and high dimensional real data, using both oracle and learned black box scores (for example, deep classifiers on images, text, and sensor data) to illustrate practical applicability.

**Additional Comments:**

Algorithm 1 is dense. A small diagram that visually explains the MCP matrix, its rows and columns and how they correspond to different hypotheses would help readers who are less comfortable with layered indices. Figure 2 partially does this, but could be integrated earlier.

**Audience:**

Yes

**Audience Explanation:**

This paper tackles a problem that a clear subset of the TMLR audience cares about: how to obtain confidence sets for changepoint locations with rigorous guarantees, rather than just point estimators or heuristic intervals.

In particular:

- The core contribution is a method that builds finite sample, distribution free confidence sets for a single changepoint under very weak assumptions (essentially exchangeability on each side). That type of set valued uncertainty quantification for changepoints is still rare, so methodologists interested in inference rather than just detection would naturally be drawn to it.

- The paper provides formal theoretical guarantees for these sets, framed within a conformal prediction style construction. Researchers working on conformal methods, selective inference, or distribution free testing would want to understand both the construction and the associated optimality result (the conformal Neyman–Pearson lemma), even if some technical details need polishing.

- The fact that the same framework can be used with black box scores from deep networks in high dimensional applications makes the results relevant to practitioners who want guarantees, not just scores or alarms, in modern ML settings.

**Claims And Evidence:**

No

**Claims Explanation:**

1. While the paper presents several compelling examples (Gaussian mean change, MNIST digit change,  CIFAR), the experimental section is largely illustrative. I would strongly recommend adding more systematic simulation studies that quantify the frequentist operating characteristics of MCP:
- In simple synthetic models (e.g. Gaussian mean/variance shifts), report empirical type I error and power of the test for $H_0^t$, as well as empirical coverage probability and average length of the resulting confidence sets, over many repetitions and across different sample sizes and signal strengths. The Gaussian mean change experiment in Appendix A could be expanded and brought into the main text for this purpose.
- Since MCP is proposed as a changepoint localization method, it would be very helpful to compare its performance on synthetic datasets against standard baselines (e.g. CUSUM based methods, binary/wild binary segmentation, kernel change point methods), in terms of localization error and, where applicable, interval width and coverage.

2. It appears there might be a typo or a misunderstanding on my part regarding the inequality presented in the analysis of the third term. Specifically, the statement that $\lim_{y \uparrow F_X^{-1}(u)} F_X(y) > u$ seems to conflict with the standard definition of the quantile function $F_X^{-1}(u)$. My understanding is that because the quantile is defined as the infimum of values where the CDF exceeds $u$, the probability accumulated strictly to the left of the quantile should naturally be less than or equal to $u$ (i.e., $\lim_{y \uparrow F_X^{-1}(u)} F_X(y) \leq u$). If this limit were strictly greater than $u$, it would imply that the threshold $u$ was crossed at an earlier point $y < F_X^{-1}(u)$, which would contradict the definition of the quantile. As this inequality is central to the deduction that $U > u$ for this case, the authors may wish to revisit this step to clarify the logic.

**Requested Changes:**

Besides providing supporting experimentation and clarifying the theoretical misunderstanding previously mentioned, I have these suggestions:

1. The related work section is long but does not clearly highlight the conceptual relationship between MCP and prior conformal prediction based approaches to changepoint or anomaly detection. Since the main novelty of the paper is a distribution free confidence set for the changepoint obtained via conformal techniques, it would be helpful to foreground this connection already in the introduction. By contrast, the classical changepoint detection literature (CUSUM, binary segmentation, WBS, multiscale methods, etc.) is well known and could be summarized more concisely, with the focus placed instead on how MCP differs from and complements the existing conformal and distribution free approaches.

2. The definition of the score functions s_rtin Section 3 is quite abstract, and later sections use the term “score” in different forms (the fully general MCP formulation vs. the simplified setting in the conformal Neyman–Pearson lemma where $s:\chi→R$. This makes it initially difficult to understand precisely what properties the score must satisfy and how the theoretical definition connects to the concrete score constructions used in the experiments. Appendix H does provide helpful concrete examples (e.g., density ratio based scores, classifier based scores), but these are somewhat hidden. I would suggest mentioning at least one of these examples immediately after the general definition in Section 3 (or explicitly pointing to Appendix H at that point), so that readers can see a concrete instance of an admissible score right away and more easily understand how the later constructions fit into the abstract framework.

3. I found the discussion around the sentence “However, we describe a method to pre test the data for exchangeability” somewhat confusing. As I understand it, the formal coverage guarantee for the confidence set $C_{1-\alpha}$ is proved under a model where a single changepoint $\xi$ is assumed to exist. In contrast, the global pre test for exchangeability in Appendix D is meant to detect the no changepoint case. It would be helpful to make precise:
- whether $H_0^t$ should be interpreted as “$\xi=t$” in a model where a changepoint is known to exist, or rather as a pure exchangeability hypothesis across the split at $t$;
- how the coverage guarantee for $C_{1-\alpha}$ should be interpreted in the no changepoint scenario, where $\xi$ is not well defined; and
- whether the optional pre test affects, even implicitly, the nominal level of the tests for $H_0^t$ or the coverage of $C_{1-\alpha}$ when a changepoint does exist.
Clarifying the precise assumptions under which the confidence set is valid, and how the pre test interacts with those assumptions, would make the theoretical guarantees easier to interpret.
- I initially found the notation around the various p values somewhat confusing, in particular the distinction between the pointwise conformal $p$ values $p_r^{(t)} $ and the aggregated $p$ values $p_t$ used to form the confidence set. While this becomes clear after careful reading, it may help readers if you explicitly separate and name these objects when they are first introduced (for example, “per point $p$ values” versus “per candidate changepoint p values”), and remind the reader which one is being used when defining the test for $H_0^t$ and the confidence set $C_{1-\alpha} $. A brief paragraph or notation table summarizing the roles of $p, p_r^{(t)}$, and $p_t$vwould greatly improve readability.

---

> ### Comment · Action_Editor_p13s · 2026-03-21
> **Reviewers did not see the response**
>
> We thank the reviewer for their helpful and constructive feedback on our work. We have uploaded a new version of our paper reflecting the below changes.
> 1. Frequentist operating characteristics of MCP
>
> We moved the Gaussian mean change experiment into the main text and augmented it with several additional aspects in order to gain insight into the operating characteristics of MCP:
>
>     We added a plot depicting power of the power of our tests for . The power of the test increases as we move further away from the true changepoint, as one would intuitively expect.
>     We ran the Gaussian mean change experiment for varying signal strengths, sample sizes, and confidence levels. We observe that the empirical coverage is very close to and that the point estimate is very small relative to the number of samples (though systematically negative, because the changepoint is in the left half).
>
> 2. Comparison against baselines
>
> As mentioned in the Related Work section, it is difficult to find appropriate baselines for the MCP algorithm because (as far as we know) it is the first to provide a distribution-free and finite-sample confidence set for the changepoint. As a result, many existing baselines are too strong to provide a good comparison, since they make far more assumptions.
>
> We have added a comparison to the parametric changepoint confidence sets described in [2], which are based on thresholding the likelihood ratio. We also added a comparison to the nonparametric MOSUM algorithm described in [1], which is based on the bootstrap.
>
> Caveats: the likelihood ratio-based method in [2] requires parametric assumptions on the data, is only asymptotically valid, and requires expensive simulations from the distribution to estimate the threshold parameters. On the other hand, MOSUM [1] is limited to univariate data with a change in mean, requires a computationally expensive bootstrap procedure, and only provides an asymptotically valid confidence interval.
>
> Given these caveats, we still compared against them. The likelihood-ratio based methods and MOSUM provide significantly smaller confidence sets than MCP in the Gaussian mean change setting, but were very non-robust, in the sense that when the parametric assumption was violated (eg: by using heavier tailed data), their coverage was significantly hurt. In contrast, the coverage of MCP is unaffected by the distributional assumptions, one of the main advantages of a distribution-free nonasymptotic guarantee.
>
> Other works on changepoint localization (such as [3]) are often only asymptotically valid and have unspecified constants which make them difficult to implement in practice. Indeed the authors of [3] themselves did not run any experiments, so we did not compare against them.
> 3. The third term in the proof of Lemma E.1 (randomized PIT)
>
> We apologize for the misunderstanding and thank the reviewer for pointing out a minor issue with our original proof of Lemma E.1. We have rectified the analysis of the third term in the lemma, and the result is still true.
> 4. Re-focusing Related Work section
>
> We have re-focused the Related Work section around the ways in which MCP differs from and complements existing conformal and distribution-free approaches to changepoint analysis.
> 5. Intuition for the score function
>
> We now reference Appendix H (which provides several concrete examples of valid score functions) immediately after Definition 3.1 (score functions), in order to provide intuition.
> 6. Pre-testing for exchangeability
>
> The test for exchangeability is a valid test that the practitioner may perform in order to check for the existence of any changepoints. However, our formal guarantees on MCP do not technically hold if the pre-test is used to decide whether or not to run MCP (due to issues surrounding post-selection inference). If MCP is always run (with or without a pre-test), then there is no post-selection inference problem and our theorems hold; the issue only arises if one runs MCP conditional on the pre-test rejecting. We have clarified this point in Appendix D.
> 7. Notation improvements
>
> We have given descriptive names to the variables
> and (which we now call per-anomaly and per-candidate p-values) and summarized their roles in a paragraph following Algorithm 1 in order to improve readability.
> 8. Improving readability of Algorithm 1
>
> We agree with the reviewer that Algorithm 1 is dense, so (as suggested) we rearrange the discussion following the algorithm to highlight its interpretation via the matrix of conformal p-values. We also move up the discussion of Figure 2, which provides some helpful intuition for Algorithm 1.
> References
>
> [1] Bootstrap confidence intervals for multiple change points based on moving sum procedures. (Cho and Kirch, 2022)
>
> [2] Confidence sets in change-point problems. (Siegmund, 1988)
>
> [3] Optimal change-point detection and localization. (Verzelen et al., 2023)

---

### Review · Reviewer_dpfL · 2026-01-07

**Summary Of Contributions:**

In this paper, the authors produce a confidence set for a single change-point using a constructed matrix of conformal p-values. The whole process is carried out under the assumption that the pre-change and post-change distributions are each exchangeable. Furthermore, the authors work under the extra, quite strong assumption that the pre-change data are independent of the post-change data. More precisely, the main contributions of the manuscript under review are:
1. The development of a new algorithm, labelled MCP, for the construction of confidence sets regarding the location of a single change-point in a given data sequence under the aforementioned assumption.
2. The confidence sets are also valid for finite-sample sizes (no strong asymptotic distributional assumptions are required).
3. The introduction of a Neyman-Pearson type result for learning the conformal score function from the data.

**Additional Comments:**

No additional comments.

**Audience:**

Yes

**Audience Explanation:**

Change-point detection is a very active area of statistical research with applications in many different areas. Developing a new algorithm for the construction of confidence sets for the location of a change-point (with possible extension to multiple change-points and multivariate data sequences) is definitely, in my opinion, something very interesting.

**Broader Impact Concerns:**

I do not have any concerns about the ethical implications of the work.

**Claims And Evidence:**

Yes

**Claims Explanation:**

The authors provide results for the performance of the proposed algorithm on a sufficient number of synthetic and real-world datasets. Indeed, the obtained confidence sets are adequately narrow, as also pointed out by the authors, to be of practical use to applied statisticians. I have some comments and concerns, though, which I would like the authors to take seriously into consideration when revising the paper. I highlight my comments in a later text box.

**Requested Changes:**

I have some comments regarding the paper, which I provide below. Even though, in my opinion, the paper's scope and its findings are interesting to the community, I strongly advise the authors to consider the comments below:
1. I am wondering how realistic the assumption is that the pre-change data are independent of the post-change data. In real-world data, this will rarely be the case; I would like the authors to comment on any pre-processing they might want to perform on the data before applying their MCP algorithm. I am asking for ways that they could use to ensure that the data under consideration satisfy (or are sufficiently close to) the assumption above.

2. I am not asking the authors to perform such an extension, but it would definitely add a lot to the paper to have the algorithm be able to cover the case of multivariate data sequences. I genuinely believe that such an extension does not require a substantial amount of tedious work.

3. The authors mention in p.3 two papers (Pettitt (1979) and Ross & Adams (2012)) about nonparametric change-point detection, which do not produce a confidence set for any possible single change-point and suffer from a lack of statistical power. However, there is the 1991 paper by Lutz Dumbgen, which actually provides results on statistical power, and, even more importantly, in Section 5 of that paper ("The asymptotic behavior of some nonparametric changepoint estimators"), the author gives confidence sets for the change-point. I would advise the authors to read that paper and revisit their statements in the relevant parts of their paper.

4. In p.5, Algorithm 1, line 9, the authors start j from r+1 and take it up to n. However, from line 8, r can obviously take the value of n. Therefore, there is a mistake in the algorithmic steps. The authors need to make sure to correct the mistake both in the paper, but also in any relevant for loops used in their code, which might also be affected by such a mistake.

5. In p.5, fourth line below the matrix of conformal values, the authors mention the $t$th row; based also on Figure 2, I think that they probably mean the $t$th column of the matrix of conformal values.

6. A suggestion for better structuring the paper (which is, nevertheless, well-written) is to put current Sections 4.1 and 4.2 before Theorem 3.1, which actually refers to those sections and their results are also used in the proof of Theorem 3.1.

7. In Algorithm 2, is $B$ equal to the sample size $n$ used earlier in the paper? Why do the authors need a separate notation?

8. In Section 4.2, the authors use the notations $s_{rt}^{(j)}$ and $s_t^{(j)}$ where $j=0,1$, I believe for the same quantity, unless I miss something.

9. The authors mention that in practice, they have found the minimum and the Bonferroni corrections to be a good choice for a combining function. I understand this, but I am wondering whether the authors could also provide (or refer the readers to) some theoretical results about such a choice.

10. In the statement of Theorem 5.1, I would advise the authors to explicitly define $\phi$, which is used in the second inequality of that theorem.

11. In Section 5, the authors estimate the likelihood ratio score function by the function $s_{rt}$ as given in the last displayed equation of that section. I would like the authors to evaluate or, at least, comment (using references or some basic calculations/simulations) on how good this estimation is for finite sample sizes. How much does someone lose in terms of accuracy or how much does such an estimation add to the computational cost of the MCP algorithm?

12. In Section 6, Line 2, the vector should start from $\xi_{k}+1$ and end at $\xi_{k+1}$. As it is now, they have two change-points in one vector.

13. In p.10, Line 5, the authors mention the popular ``Isolate-Detect'' paradigm and give as a relevant reference Section 5.2 in Truong et al. (2020), which is not the case. That section is related to approximate detection of change-points with special mention to window-based methods, binary segmentation, and bottom-up approaches. Regarding the ``Isolate-Detect'' paradigm, the authors should refer to the relevant Anastasiou & Fryzlewicz (2022) paper.

14. In p.11, Line -3, the authors mention that, due to their Theorem 3.1, the true coverage probability is {\textbf{exactly}} 50% and 95%. However, looking at the statement of the theorem, and unless I am missing something, I would say that the coverage probability is {\textbf{at least}} 50% and 95%.

15. In p.13, Line 2, the authors give the consistency rate for the point estimator. Is this rate optimal? Under what assumptions and for what type of changes?

16. In p.13, Lines 3-4, the authors mentioned that, due to the true change-point being in the left half of the data, the point estimator tends to be biased towards the left of the true change-point. I do not understand why this is expected or reasonable.

17. I have noticed that both in the synthetic data and also in the real-world data examples, the true changepoint is at $\xi = 400$. Why is this the case? Is there anything special about that value? I would advise the readers to provide results when the true change-point is at different locations. Also, what happens if you put the change-point near the edges of the set from 1 up to $n$?

18. In p.16, Line -7, the authors mention that ``most prior nonparametric changepoint localization focus on localizing a change in mean''. This is not really the case; there is an extensive literature (Carlstein (1988), Dumbgen (1991), Lee (1996), Zou et al. (2014), and many others) of non-parametric change-point detection for any kind of changes (two different distributions before and after the change-point, with the mean possibly remaining the same).

19. In Section 7.4, the authors mention that $\xi = 300$, but in Figure 11 it seems that $\xi = 400$ again (even though in the caption $\xi = 300$ is mentioned). Furthermore, the label, the caption, and the graph itself are again misleading ($\xi = 300$ vs $\xi = 400$) for Figure 12. I would like the authors to clarify this.

20. In p.20, first line of Appendix B, the labelling is wrong since the MNIST digit change experiment is in Appendix A and not in Section 7.

---

> ### Comment · Action_Editor_p13s · 2026-03-21
> **Reviewers did not see the response**
>
> We thank the reviewer for their helpful feedback. We have uploaded a new version reflecting the below changes.
>
>     The assumption that the pre- and post-change data are independent is hard to verify since the location of the change is unknown. However, this is a very common assumption in the changepoint literature and the distribution shift literature, including most/all other papers that we cite (including point 3).
>     Our algorithm places no restrictions on the type of input data, which is one of its features. Eg: the CIFAR-100 experiment involves image data (which is multivariate); the score function is constructed using a classifier. With access to a classifier or density estimator, Appendix H already outlines how our algorithm can be used.
>     Carlstein (1988), Lee (1996), and Zou (2014) only provide consistent point estimators, not confidence sets. Dumbgen (1991) produces confidence sets (via an expensive bootstrap), but only asymptotically valid, and relying on several complex theoretical assumptions which are difficult to verify in practice, unlike our nonasymptotic guarantee under only the assumption discussed in point 1. When instantiated to an exponential family, Dumbgen has an explicit formula for the confidence set matching Siegmund (1988), which we compare to in Section 7.1.2. We thank the reviewer for bringing our attention to these works and have updated the Related Work accordingly.
>     We rectified this typo; this error was not in the code.
>     Yes, we intended to say "th column", now fixed.
>     As suggested, we moved Thm 3.1 after Sec 4.1, 4.2.
>     is not necessarily equal to , and it is a user-chosen parameter. We clarify this in the inputs of Algorithms 2, 3.
>     We fixed this typo.
>     One reason why the minimum works well is because (for eg) to the right of the changepoint, the right p-value is exactly uniformly distributed and we expect the left p-value to be small. In general, one of the left and right p-values does not contribute any power. Therefore, their minimum makes sense (transformed so the result remains a p-value).
>     We now define in the statement of Theorem 5.1.
>     The followup work of Bhattacharya and Ramdas (2025, Section 7) shows that under light assumptions, using kernel density estimators for the likelihood ratio in MCP results in a near-optimal score, by controlling
>     , where is the conformal p-value constructed using score , is the optimal likelihood ratio score, and
>     is the score using density estimators. An analysis like this for a classifier-based score as in Section 5 is still open, but plausible. In terms of computation, computing the score in the way we outlined does not incur any additional cost over using the oracle score, since we only need to perform a single pass at the start to compute the most common classes in the prefix and postfix arrays. We now note this in Section 5.
>     We have fixed the start location of the vector.
>     We now reference Anastasiou and Fryzlewicz (2022) for "Isolate-Detect".
>     We modified this statement as suggested.
>     If is unclear whether the consistency rate of our point estimator is optimal. The followup work of Bhattacharya and Ramdas (2025) derived this rate in a rather general setup, but that paper did not derive lower bounds, and it is the first distribution-free upper bound we are aware of.
>     We cannot explain why the point estimator may be slightly biased to the left when the true change is in the left half of the data. It makes sense that such a phenomenon could occur, because (if , ) the left p-value at has access to 400 exchangeable points and 5 anomalies and the right p-value at has access to 600 exchangeable points and 5 anomalies, but the complex dependence between these p-values prevents us from giving a complete explanation. We noted the phenomenon in our experiments, and simply alerted the reader to it, though it does not affect validity of our confidence sets.
>     There is nothing special about . We added Gaussian mean change experiments in Section 7.1 where the true changepoint is and . Although we expect the power of our method to improve when the changepoint is not near the boundaries, we observe in practice that the size of the confidence set does not become too much larger and our method is still practical even when the true changepoint is near the edges.
>     We have updated this text to say only that much prior work focuses on localizing a change in mean (CUSUM-type methods), and these methods are not applicable in our setting. For a further discussion of the references provided by the reviewer, please see our answer to point 3.
>     We have rectified the sample image in the CIFAR-100 experiment to have a changepoint at , for consistency with the following figures.
>     We have fixed the cross-reference to the MNIST digit change experiment.

---

### Review · Reviewer_HNe2 · 2026-02-13

**Summary Of Contributions:**

This work proposes MCP (matrix of conformal p-values), a method for offline changepoint localization method that outputs a confidence set for a single changepoint under the assumption that each segment is exchangeable. They compute candidate-wise p-values by aggregating conformal “per-anomaly” p-values on both sides of a candidate split, and invert these tests to form a confidence set.

The main contributions are:
- A finite-sample coverage guarantee for the changepoint confidence set constructed by MCP
- A conformal Neyman–Pearson–style optimality result motivating likelihood-ratio / classifier-based score choices
- They also provide experiments on synthetic and real-world sequences, including using black-box pretrained classifiers on images/text/sensors

Overall, the core contribution is interesting and nontrivial: a finite-sample, assumption-lean confidence set for a changepoint location via conformal inversion with practical experiments

**Audience:**

Yes

**Audience Explanation:**

TMLR readers interested in conformal inference, distribution-free methods, and changepoint analysis would be interested in a method that offers finite-sample valid changepoint confidence sets under minimal assumptions.
The paper also shows how to leverage pretrained classifiers as score functions for high-dimensional sequences (images/text/sensors).

**Claims And Evidence:**

Yes

**Claims Explanation:**

The main theoretical result (finite-sample coverage for the MCP confidence set under exchangeability) is proven as Theorem 4.1.

The score-function guidance result is backed by Theorem 5.1 and motivates the classifier-based constructions used later.

Empirically, the paper provides synthetic studies showing robustness where classical mean-change CIs break (e.g., Cauchy mean-change where baselines fail badly while MCP retains reasonable coverage/width). They also provide real-world experiments using pretrained models.

One limitation is that the confidence-set construction is explicitly not valid when there is no changepoint without a separate pre-test.

**Requested Changes:**

- The paper defines “no change” via $\zeta = n$ but later states that the Neyman-inverted confidence set is “not valid if there is no changepoint” and points to a pre-test in Appendix D. What is the target parameter/guarantee when $\zeta = n$, since $P_1$ is effectively empty? If you recommend an optional pre-test for global exchangeability, does it change any nominal coverage/type-I guarantees when a changepoint does exist?

- If possible, add a computational complexity / scalability analysis or discussion. Algorithm 1 has nested loops over $(t, r, j)$ and Algorithm 2 adds Monte Carlo calibration with $B$ simulated runs. It'd be nice to know what the time and memory complexities are, as well as measured runtime vs. sequence length. I'd also be curious about recommended fast variants (empirical/asymptotic) vs. permutation testing and what can be cached/reused.

- Strengthen experimental reporting of coverage. In simulation tables, please report uncertainty (e.g., binomial standard errors / CIs across repetitions). You report empirical coverage in Table 1 (with uncertainty annotations in the current draft), and some entries are visibly below nominal (e.g., 0.93 for a nominal 0.95 case). Please (i) consistently report uncertainty across simulation tables/figures (SEs or CIs over repetitions), and (ii) briefly diagnose whether deviations are Monte Carlo error (finite $B$ and/or finite simulation reps), implementation details, or due to something else.

- Ablations on score functions and p-value combining. Since you discuss independence of left/right p-values for non-adaptive scores and recommend Bonferroni under dependence, an ablation would be informative. Could the authors compare minimum/Fisher/Bonferroni (or others you consider) across (a) non-adaptive vs. adaptive score families and (b) settings with induced dependence, reporting the conservativeness/width/power tradeoffs.

---

> ### Author Response · Authors · 2026-02-18
>
> We thank the reviewer for their helpful feedback. We have uploaded a new version reflecting the below changes.
>
> 1. We found a clean way to remove the pre-test altogether. Instead of only testing $\mathcal{H}\_{0t}$ for $t<n$, the MCP algorithm now tests them for $t \leq n$, performing the Neyman inversion to produce the confidence set. Now, the matrix of p-values is size $n \times n$, and not $n \times (n-1)$ like it was earlier, which is also more aesthetically pleasing. Then, if $n$ is included in the final confidence set, the practitioner may conclude that there is a possibility that no change occurred. The advantage of this approach is that since we are not running the MCP algorithm conditional on a pre-test, our validity guarantees continue to hold and the interpretation of our confidence set becomes much simpler. Since the test for $\mathcal{H}_{0n}$ is typically very powerful, as shown in Section 7.1, the resulting confidence sets are unaffected in the case that $\xi \neq n$. On the other hand, if $\xi = n$, the practitioner will see significantly wider confidence sets, but which include $n$; we show a relevant experiment in Appendix E.
> 2. The computational complexity of the MCP algorithm is dependent on the choice of score function, and the base algorithm runs in $O(n^2)$ time. In fact, most of the score functions that we suggest in Section 5 and Appendix I can be computed with an $O(n^2)$ pre-processing step, so our algorithm typically has an overall quadratic complexity. This is reflected in the implementations provided in the supplementary material. Performing the empirical test technically adds $O(Bn)$ to the time complexity, but is typically very fast in practice due to existence of pre-computed quantiles for the KS test in software packages. We have updated the text of the paper to include this discussion.
> 3. The reason that our empirical coverage is sometimes lower than the nominal coverage is due to Monte Carlo error. As noted, sometimes the empirical coverage that we reported is 0.93 when the nominal coverage is 0.95, but the nominal coverage is always within the binomial errors that we report in the paper. We have updated the paper to include standard errors for all other reported statistics (width, absolute deviation, and bias).
> 4. We now compare between different combining methods (minimum, Fisher, and Bonferroni) in Appendix C, across the oracle and KDE scores in the Gaussian mean change setting. Since the KDE score function is adaptive and thereby induces dependence between the left and right p-values, our theoretical results are technically only valid if one uses the Bonferroni combining method. However, we observe that in practice, the choice of combining method does not matter much in terms of the resulting coverage and width obtained.

---

### Decision · Action_Editor_p13s · 2026-04-13

**Recommendation:** Accept as is

**Audience:**

Yes

**Audience Explanation:**

Yes, all three reviewers are aligned.

**Claims And Evidence:**

Yes

**Claims Explanation:**

The reviewers are satisfied that both theoretical and synthetic data results back the claims in the paper. While some reviewers would like to have seen more demonstrative real-world examples, it does not detract from providing evidence towards the main claims in the paper. After an OpenReview configuration mishap, two reviewers have responded positively to the authors' feedback. I reviewed the third respons and find it adequate.